# Evaluations and Methods for Explanation through Robustness Analysis

**Cheng-Yu Hsieh**[1*]**, Chih-Kuan Yeh**[2]**, Xuanqing Liu**[3]**, Pradeep Ravikumar**[2]**,**
**Seungyeon Kim**[4]**, Sanjiv Kumar**[4]**, Cho-Jui Hsieh**[3,4]

[1]Paul G. Allen School of Computer Science, University of Washington
[2]Machine Learning Department, Carnegie Mellon University
[3]Department of Computer Science, UCLA
[4]Google Research
`cydhsieh@cs.washington.edu`,`{cjyeh, pradeepr}@cs.cmu.edu`,
`xqliu@cs.ucla.edu`,`{seungyeonk, sanjivk}@google.com`,
`chohsieh@cs.ucla.edu`

## Abstract

*Feature based explanations*, that provide importance of each feature towards the model prediction, is arguably one of the most intuitive ways to explain a model. In this paper, we establish a novel set of evaluation criteria for such feature based explanations by robustness analysis. In contrast to existing evaluations which require us to specify some way to "remove" features that could inevitably introduces biases and artifacts, we make use of the subtler notion of smaller adversarial perturbations. By optimizing towards our proposed evaluation criteria, we obtain new explanations that are loosely necessary and sufficient for a prediction. We further extend the explanation to extract the set of features that would move the current prediction to a target class by adopting targeted adversarial attack for the robustness analysis. Through experiments across multiple domains and a user study, we validate the usefulness of our evaluation criteria and our derived explanations.

## 1    Introduction

There is an increasing interest in machine learning models to be credible, fair, and more generally *interpretable* (Doshi-Velez & Kim, 2017). Researchers have explored various notions of model interpretability, ranging from trustability (Ribeiro et al., 2016), fairness of a model (Zhao et al., 2017), to characterizing the model's weak points (Koh & Liang, 2017; Yeh et al., 2018). Even though the goals of these various model interpretability tasks vary, the vast majority of them use so called *feature based explanations*, that assign importance to individual features (Baehrens et al., 2010; Simonyan et al., 2013; Zeiler & Fergus, 2014; Bach et al., 2015; Ribeiro et al., 2016; Lundberg & Lee, 2017; Ancona et al., 2018; Sundararajan et al., 2017; Zintgraf et al., 2017; Shrikumar et al., 2017; Chang et al., 2019). There have also been a slew of recent *evaluation* measures for feature based explanations, such as completeness (Sundararajan et al., 2017), sensitivity-n (Ancona et al., 2018), infidelity (Yeh et al., 2019), causal local explanation metric (Plumb et al., 2018), and most relevant to the current paper, removal- and preservation-based criteria (Samek et al., 2016; Fong & Vedaldi, 2017; Dabkowski & Gal, 2017; Petsiuk et al., 2018). A common thread in all these evaluation measures is that for a good feature based explanation, the most salient features are necessary, in that removing them should lead to a large difference in prediction score, and are also sufficient in that removing non-salient features should not lead to a large difference in prediction score.

Thus, common evaluations and indeed even methods for feature based explanations involve measuring the function difference after "removing features", which in practice is done by setting the feature value to some reference value (also called baseline value sometimes). However, this would favor feature values that are far way from the baseline value (since this corresponds to a large perturbation, and hence is likely to lead to a function value difference), causing an intrinsic bias for these methods and evaluations. For example, if we set the feature value to black in RGB images, this introduces a bias favoring bright pixels: explanations that optimize such evaluations often omit important dark objects such as a dark-colored dog. An alternative approach to "remove features" is to sample from

---

*This work was done when the author was visiting UCLA.

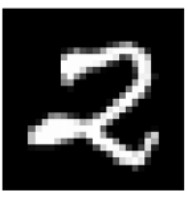 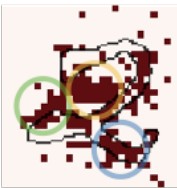

Figure 1: Illustration of our explanation highlighting both pertinent positive and negative features that support the prediction of "2". The blue circled region corresponds to pertinent positive features that when its value is perturbed (from white to black) will make the digit resemble "7"; while the green and yellow circled region correspond to pertinent negative features that when turned on (black to white) will shape the digit into "0","8", or "9".

some predefined distribution or a generative model (Chang et al., 2019). This nevertheless in turn incurs the bias inherent to the generative model, and accurate generative models that approximate the data distribution well might not be available in all domains.

In this work, instead of defining prediction changes with "removal" of features (which introduces biases as we argued), we alternatively consider the use of small but *adversarial perturbations*. It is natural to assume that adversarial perturbations on irrelevant features should be ineffective, while those on relevant features should be effective. We can thus measure the necessity of a set of relevant features, provided by an explanation, by measuring the consequences of adversarially perturbing their feature values: if the features are indeed relevant, this should lead to an appreciable change in the predictions. Complementarily, we could measure the sufficiency of the set of relevant features via measuring consequences of adversarially perturbing its complementary set of irrelevant features: if the perturbed features are irrelevant, this should not lead to an appreciable change in the predictions. We emphasize that by our definition of "important features", our method may naturally identify both pertinent positive and pertinent negative features (Dhurandhar et al., 2018) since both pertinent positive and pertinent negative features are the most susceptible to adversarial perturbations, and we demonstrate the idea in Figure 1. While exactly computing such an effectiveness measure is NP-hard (Katz et al., 2017), we can leverage recent results from test-time robustness (Carlini & Wagner, 2017; Madry et al., 2017), which entail that perturbations computed by adversarial attacks can serve as reasonably tight upper bounds for our proposed evaluation. Given this adversarial effectiveness evaluation measure, we further design feature based explanations that optimize this evaluation measure.

To summarize our contributions:

- We define new evaluation criteria for feature based explanations by leveraging robustness analysis involving small adversarial perturbations. These reduce the bias inherent in other recent evaluation measures that focus on "removing features" via large perturbations to some reference values, or sampling from some reference distribution.
- We design efficient algorithms to generate explanations that optimize the proposed criteria by incorporating game theoretic notions, and demonstrate the effectiveness and interpretability of our proposed explanation on image and language datasets: via our proposed evaluation metric, additional objective metrics, as well as qualitative results and a user study.[1]

## 2 RELATED WORK

Our work defines a pair of new objective evaluation criteria for feature based explanations, where existing measurements can be roughly categorized into two families. This first family of explanation evaluations are based on measuring fidelity of the explanation to the model. Here, the feature based explanation is mapped to a simplified model, and the fidelity evaluations measure how well this simplified model corresponds to the actual model. A common setting is where the feature vectors are locally binarized at a given test input, to indicate presence or "removal" of a feature. A linear model with the explanation weights as coefficients would then equal to the sum of attribution values for all present features. *Completeness* or *Sum to Delta* requires the sum of all attributions to equal the prediction difference between the original input and a baseline input (Sundararajan et al., 2017; Shrikumar et al., 2017), while *Sensitivity-n* further generalize this to require the sums of subsets of attribution values to equal the prediction difference of the input with features present or absent corresponding to the subset, and the baseline (Ancona et al., 2018). *Local accuracy* (Ribeiro et al., 2016; Lundberg & Lee, 2017) measures the fidelity of local linear regression model corresponding to explanation weights; while *Infidelity* is a framework that encompasses these instances above (Yeh

---

[1]Code available at `https://github.com/ChengYuHsieh/explanation_robustness`.

et al., 2019). The second family of explanation evaluations are removal- and preservation-based measurements (Samek et al., 2016; Fong & Vedaldi, 2017; Dabkowski & Gal, 2017; Petsiuk et al., 2018), which evaluate the ranking of feature attribution score, with the key insight that removing the most (least) salient features should lead to the most (least) significant function value difference. Unlike previous evaluations, our proposed evaluation criteria are based on more nuanced and smaller adversarial perturbations which do not rely on operationalizing "feature removal".

Another close line of research is counterfactual and contrastive explanations (Wachter et al., 2017; Dhurandhar et al., 2018; Hendricks et al., 2018; van der Waa et al., 2018; Chang et al., 2019; Goyal et al., 2019; Joshi et al., 2019; Poyiadzi et al., 2020), which answer the question: "what to alter in the current input to change the outcome of the model". These studies enable interpretation of a model decision through a contrastive perspective and facilitate actionable means that one can actually undertake to change the model prediction (Joshi et al., 2019; Poyiadzi et al., 2020). While "the closest possible world" provided by counterfactual explanations sheds light on the features that are "necessary" for the prediction, it could fail to identify relevant features that are "sufficient" for the current prediction. Xu et al. (2018) add group sparsity regularization to adversarial attacks that improves the semantic structure of the adversarial perturbations. Instead of outputting general adversarial perturbations, our work focuses specifically on identifying necessary and sufficient relevant features by measuring their susceptibility to adversarial perturbations. Ribeiro et al. (2018) find a set of features that once fixed, the prediction does not change much with high probability when perturbing other features. They measure this via a sampling approach, which could be infeasible for high-dimensional images without super-pixels. In contrast, our method considers the class of adversarial perturbations, which is much more computationally feasible compared to sampling high-dimensional spaces. Guidotti et al. (2019) exploit latent feature space to learn local decision rules with image exemplars that support the current prediction and counter-exemplars that contrast the current prediction. While the explanation is capable of highlighting both areas pertinent to the prediction and areas that pushes the prediction towards another class, the method relies on an effective generative model which might not always be available in different applications.

## 3 ROBUSTNESS ANALYSIS FOR EVALUATING EXPLANATIONS

### 3.1 PROBLEM NOTATION

We consider the setting of a general $K$-way classification problem with input space $\mathcal{X} \subseteq \mathbb{R}^d$, output space $\mathcal{Y} = \{1, \ldots, K\}$, and a predictor function $f : \mathcal{X} \to \mathcal{Y}$ where $f(\boldsymbol{x})$ denotes the output class for some input example $\boldsymbol{x} = [\boldsymbol{x}_1, \ldots, \boldsymbol{x}_d] \in \mathcal{X}$. Then, for a particular prediction $f(\boldsymbol{x}) = y$, a common goal of feature based explanations is to extract a compact set of relevant features with respect to the prediction. We denote the set of relevant features provided by an explanation as $S_r \subseteq U$ where $U = \{1, \ldots, d\}$ is the set of all features, and use $\overline{S_r} = U \setminus S_r$, the complementary set of $S_r$, to denote the set of irrelevant features. We further use $\boldsymbol{x}_S$ to denote the features within $\boldsymbol{x}$ that are restricted to the set $S$.

### 3.2 EVALUATION THROUGH ROBUSTNESS ANALYSIS

A common thread underlying evaluations of feature based explanations (Samek et al., 2016; Petsiuk et al., 2018), even ranging over axiomatic treatments (Sundararajan et al., 2017; Lundberg & Lee, 2017), is that the importance of a set of features corresponds to the change in prediction of the model when the features are removed from the original input. Nevertheless, as we discussed in previous sections, operationalizing such a removal of features, for instance, by setting them to some reference value, introduces biases (see Appendix A and Section 5.2 for formal discussion and empirical results on the impact of reference values). To finesse this, we leverage adversarial robustness, but to do so in this context, we rely on two key intuitive assumptions that motivate our method:

**Assumption 1:** When the values of the important features are anchored (fixed), perturbations restricted to the complementary set of features has a weaker influence on the model prediction.

**Assumption 2:** When perturbations are restricted to the set of important features, fixing the values of the rest of the features, even small perturbations could easily change the model prediction.

Based on these two assumptions, we propose a new framework leveraging the notion of adversarial robustness on feature subsets, as defined below, to evaluate feature based explanations.

**Definition 3.1** *Given a model $f$, an input $\boldsymbol{x}$, and a set of features $S \subseteq U$ where $U$ is the set of all features, the minimum adversarial perturbation norm on $\boldsymbol{x}_S$, which we will also term Robustness-S of $\boldsymbol{x}$ is defined as:*

$$\epsilon^*_{\boldsymbol{x}_S} = g(f, \boldsymbol{x}, S) = \left\{ \min_{\boldsymbol{\delta}} \|\boldsymbol{\delta}\|_p \text{ s.t. } f(\boldsymbol{x} + \boldsymbol{\delta}) \neq y, \, \boldsymbol{\delta}_{\overline{S}} = 0 \right\}, \tag{1}$$

*where $y = f(\boldsymbol{x})$, $\overline{S} = U \setminus S$ is the complementary set of features, and $\boldsymbol{\delta}_{\overline{S}} = 0$ means that the perturbation is constrained to be zero along features in $\overline{S}$.*

Suppose that a feature based explanation partitions the input features of $\boldsymbol{x}$ into a relevant set $S_r$, and an irrelevant set $\overline{S_r}$, Assumption 1 implies that the quality of the relevant set can be measured by $\epsilon^*_{\boldsymbol{x}_{\overline{S_r}}}$ – by keeping the relevant set unchanged, and measuring the adversarial robustness norm by perturbing only the irrelevant set. Specifically, from Assumption 1, a larger coverage of pertinent features in set $S_r$ entails a higher robustness value $\epsilon^*_{\boldsymbol{x}_{\overline{S_r}}}$. On the other hand, from Assumption 2, a larger coverage of pertinent features in set $S_r$ would in turn entail a smaller robustness value $\epsilon^*_{\boldsymbol{x}_{S_r}}$, since only relevant features are perturbed. More formally, we propose the following twin criteria for evaluating the quality of $S_r$ identified by any given feature based explanation.

**Definition 3.2** *Given an input $\boldsymbol{x}$ and a relevant feature set $S_r$, we define Robustness-$\overline{S_r}$ and Robustness-$S_r$ of the input $\boldsymbol{x}$ as the following:*

$$\text{Robustness-}\overline{S_r} = \epsilon^*_{\boldsymbol{x}_{\overline{S_r}}}. \qquad \text{Robustness-}S_r = \epsilon^*_{\boldsymbol{x}_{S_r}}.$$

Following our assumptions, a set $S_r$ that has larger coverage of relevant features would yield *higher* Robustness-$\overline{S_r}$ and *lower* Robustness-$S_r$.

**Evaluation for Feature Importance Explanations.** While Robustness-$\overline{S_r}$ and Robustness-$S_r$ are defined on sets, general feature attribution based explanations could also easily fit into the evaluation framework. Given any feature attribution method that assigns importance score to each feature, we can sort the features in descending order of importance weights, and provide the top-$K$ features as the relevant set $S_r$. The size of $K$ (or $|S_r|$), can be specified by the users based on the application. An alternative approach that we adopt in our experiments is to vary the size of set $K$ and plot the corresponding values of Robustness-$\overline{S_r}$ and Robustness-$S_r$ over different values of $K$. With a graph where the $X-$axis is the size of $K$ and the $Y-$axis is Robustness-$\overline{S_r}$ or Robustness-$S_r$, we are then able to plot an evaluation curve for an explanation and in turn compute its the area under curve (AUC) to summarize its performance. A larger (smaller) AUC for Robustness-$\overline{S_r}$ (Robustness-$S_r$) indicates a better feature attribution ranking. Formally, given a curve represented by a set of points $\mathcal{C} = \{(x_0, y_0), \dots, (x_n, y_n)\}$ where $x_{i-1} < x_i$, we calculate the AUC of the curve by: $AUC(\mathcal{C}) = \sum_{i=1}^n (y_i + y_{i-1})/2 * (x_i - x_{i-1})$.

**Relation to Insertion and Deletion Criteria.** We relate the proposed criteria to a set of commonly adopted evaluation metrics: the *Insertion* and *Deletion* criteria (Samek et al., 2016; Petsiuk et al., 2018; Sturmfels et al., 2020). The Insertion score measures the model's function value when only the top-relevant features, given by an explanation, are presented in the input while the others are removed (usually by setting them to some reference value representing feature missingness). The Deletion score, on the other hand, measures the model's function value when the most relevant features are masked from the input. As in our evaluation framework, we could plot the evaluation curves for Insertion (Deletion) score by progressively increasing the number of top-relevant features. A larger (smaller) AUC under Insertion (Deletion) then indicates better explanation, as the identified relevant features could indeed greatly influence the model prediction. We note that optimizing the proposed Robustness-$\overline{S_r}$ and Robustness-$S_r$ could roughly be seen as optimizing a lower bound for the Insertion and Deletion score respectively. This follows from the intuition: Robustness-$\overline{S_r}$ considers features that when anchored, would make the prediction most robust to "adversarial perturbation". Since adversarial perturbation is the worst case of "any arbitrary perturbations", the prediction will also be robust to different removal techniques (which essentially correspond to different perturbations) considered in the evaluation of Insertion score; The same applies to the connection between Robustness-$S_r$ and Deletion score. We shall see in the experiment section that explanation optimizing our robustness measurements enjoys competitive performances on the Insertion / Deletion criteria.

**Untargeted v.s. Targeted Explanation.** Definition 3.1 corresponds to the untargeted adversarial robustness – a perturbation that changes the predicted class to any label other than $y$ is considered as a successful attack. Our formulation can also be extended to **targeted adversarial robustness**, where we replace Eqn. 1 by:

$$\epsilon^*_{\boldsymbol{x}_S,t} = \left\{ \min_{\boldsymbol{\delta}} \|\boldsymbol{\delta}\|_p \text{ s.t. } f(\boldsymbol{x} + \boldsymbol{\delta}) = t; \boldsymbol{\delta}_{\overline{S}} = 0 \right\}, \tag{2}$$

where $t$ is the targeted class. Using this definition, our approach will try to address the question "Why is this example classified as $y$ instead of $t$" by highlighting the important features that contrast between class $y$ and $t$. Further examples of the "targeted explanations" are in the experiment section.

**Robustness Evaluation on Feature Subset.** It is known that computing the exact minimum distortion distance in modern neural networks is intractable (Katz et al., 2017), so many different methods have been developed to estimate the value. Adversarial attacks, such as C&W (Carlini & Wagner, 2017) and projected gradient descent (PGD) attack (Madry et al., 2017), aim to find a feasible solution of Eqn. 1, which leads to an upper bound of $\epsilon^*_{\boldsymbol{x}_S}$. They are based on gradient based optimizers which are usually efficient. On the other hand, neural network verification methods aim to provide a lower bound of $\epsilon^*_{\boldsymbol{x}_S}$ to ensure that the model prediction will not change within certain perturbation range (Singh et al., 2018; Wong & Kolter, 2018; Weng et al., 2018; Gehr et al., 2018; Zhang et al., 2018; Wang et al., 2018; Zhang et al., 2019). The proposed framework can be combined with any method that aims to approximately compute Eqn. 1, including attack, verification, and some other statistical estimations (see Appendix B for more discussions on estimating adversarial robustness for different types of model). However, for simplicity we only choose to evaluate Eqn. 1 by the state-of-the-art PGD attack (Madry et al., 2017), since the verification methods are too slow and often lead to much looser estimation as reported in some recent studies (Salman et al., 2019). Our additional constraint restricting perturbation to only be on a subset of features specifies a set that is simple to project onto, where we set the corresponding coordinates to zero at each step of PGD.

## 4 EXTRACTING RELEVANT FEATURES THROUGH ROBUSTNESS ANALYSIS

Our adversarial robustness based evaluations allow us to evaluate any given feature based explanation. Here, we set out to design new explanations that explicitly optimize our evaluation measure. We focus on feature set based explanations, where we aim to provide an important subset of features $S_r$. Given our proposed evaluation measure, an optimal subset of feature $S_r$ would aim to maximize (minimize) Robustness-$\overline{S_r}$ (Robustness-$S_r$), under a cardinality constraint on the feature set, leading to the following set of optimization problems:

$$\underset{S_r \subseteq U}{\text{maximize}} \ g(f, \boldsymbol{x}, \overline{S_r}) \ \text{ s.t. } \ |S_r| \leq K \tag{3}$$

$$\underset{S_r \subseteq U}{\text{minimize}} \ g(f, \boldsymbol{x}, S_r) \ \text{ s.t. } \ |S_r| \leq K \tag{4}$$

where $K$ is a pre-defined size constraint on the set $S_r$, and $g(f, \boldsymbol{x}, S)$ computes the the minimum adversarial perturbation from Eqn. 1, with set-restricted perturbations.

It can be seen that this sets up an adversarial game for Eqn. 3 (or a co-operative game for Eqn. 4). In the adversarial game, the goal of the feature set explainer is to come up with a set $S_r$ such that the minimal adversarial perturbation is as large as possible, while the adversarial attacker, given a set $S_r$, aims to design adversarial perturbations that are as small as possible. Conversely in the co-operative game, the explainer and attacker cooperate to minimize the perturbation norm. Directly solving these problems in Eqn. 3 and Eqn. 4 is thus challenging, which is exacerbated by the discrete input constraint that makes it intractable to find the optimal solution. We therefore propose a greedy algorithm in the next section to estimate the optimal explanation sets.

### 4.1 GREEDY ALGORITHM TO COMPUTE OPTIMAL EXPLANATIONS

We first consider a greedy algorithm where, after initializing $S_r$ to the empty set, we iteratively add to $S_r$ the most promising feature that optimizes the objective at each local step until $S_r$ reaches the size constraint. We thus sequentially solve the following sub-problem at every step $t$:

$$\arg\max_i \ g(f, \boldsymbol{x}, \overline{S_r^t \cup i}), \ \text{ or } \ \arg\min_i \ g(f, \boldsymbol{x}, S_r^t \cup i), \ \forall i \in \overline{S_r^t} \tag{5}$$

where $S_r^t$ is the relevant set at step $t$, and $S_r^0 = \emptyset$. We repeat this subprocedure until the size of set $S_r^t$ reaches $K$. A straightforward approach for solving Eqn. 5 is to exhaustively search over every

single feature. We term this method **Greedy**. While the method eventually selects $K$ features for the relevant set $S_r$, it might lose the sequence in which the features were selected. One approach to encode this order would be to output a feature explanation that assigns higher weights to those features selected earlier in the greedy iterations.

## 4.2 GREEDY BY SET AGGREGATION SCORE

The main downside of using the greedy algorithm to optimize the objective function is that it ignores the interactions among features. Two features that may seem irrelevant when evaluated separately might nonetheless be relevant when added simultaneously. Therefore, in each greedy step, instead of considering how each individual feature will marginally contribute to the objective $g(\cdot)$, we propose to choose features based on their expected marginal contribution when added to the union of $S_r$ and a random subset of unchosen features. To measure such an aggregated contribution score, we draw from cooperative game theory literature (Dubey & Shapley, 1979; Hammer & Holzman, 1992) to reduce this to a linear regression problem. Formally, let $S_r^t$ and $\overline{S_r^t}$ be the ordered set of chosen and unchosen features at step $t$ respectively, and $\mathcal{P}(\overline{S_r^t})$ be all possible subsets of $\overline{S_r^t}$. We measure the expected contribution that including each unchosen feature to the relevant set would have on the objective function by learning the following regression problem:

$$\boldsymbol{w}^t, c^t = \arg\min_{\boldsymbol{w}, c} \sum_{S \in \mathcal{P}(\overline{S_r^t})} ((\boldsymbol{w}^T b(S) + c) - v(S_r^t \cup S))^2, \tag{6}$$

where $b : \mathcal{P}(\overline{S_r^t}) \to \{0, 1\}^{|\overline{S_r^t}|}$ is a function that projects a set into its corresponding binary vector form: $b(S)[j] = \mathbb{I}(\overline{S_r^t}[j] \in S)$, and $v(\cdot)$ is set to be the objective function in Eqn. 3 or Eqn. 4: $v(S_r) = g(f, \boldsymbol{x}, \overline{S_r})$ for optimizing Eqn. 3; $v(S_r) = g(f, \boldsymbol{x}, S_r)$ for optimizing Eqn. 4. We note that $\boldsymbol{w}^t$ corresponds to the well-known Banzhaf value (Banzhaf III, 1964) when $S_r^t = \emptyset$, which can be interpreted as the importance of each player by taking coalitions into account (Dubey & Shapley, 1979). Hammer & Holzman (1992) show that the Banzhaf value is equivalent to the optimal solution of linear regression with pseudo-Boolean functions as targets, which corresponds to Eqn. 6 with $S_r^t = \emptyset$. At each step $t$, we can thus treat the linear regression coefficients $\boldsymbol{w}^t$ in Eqn. 6 as each corresponding feature's expected marginal contribution when added to the union of $S_r$ and a random subset of unchosen features.

We thus consider the following set-aggregated variant of our greedy algorithm in the previous section, which we term **Greedy-AS**. In each greedy step $t$, we choose features that are expected to contribute most to the objective function, i.e. features with highest (for Eqn. 3) or lowest (for Eqn. 4) aggregation score (Banzhaf value), rather than simply the highest marginal contribution to the objective function as in vanilla greedy. This allows us to additionally consider the interactions among the unchosen features when compared to vanilla greedy. The chosen features each step are then added to $S_r^t$ and removed from $\overline{S_r^t}$. When $S_r^t$ is not $\emptyset$, the solution of Eqn. 6 can still be seen as the Banzhaf value where the players are the unchosen features in $\overline{S_r^t}$, and the value function computes the objective when a subset of players is added into the current set of chosen features $S_r^t$. We solve the linear regression problem in Eqn. 6 by sub-sampling to lower the computational cost, and we validate the effectiveness of Greedy and Greedy-AS in the experiment section. [2]

## 5 EXPERIMENTS

In this section, we first evaluate different model interpretability methods on the proposed criteria. We justify the effectiveness of the proposed Greedy-AS. We then move onto further validating the benefits of the explanations extracted by Greedy-AS through comparisons to various existing methods both quantitatively and qualitatively. Finally, we demonstrate the flexibility of our method with the ability to provide targeted explanations as mentioned in Section 3.2. We perform the experiments on two image datasets, MNIST LeCun et al. (2010) and ImageNet (Deng et al., 2009), as well as a text classification dataset, Yahoo! Answers (Zhang et al., 2015). On MNIST, we train a convolutional neural network (CNN) with 99% testing accuracy. On ImageNet, we deploy a pre-trained ResNet model obtained from the Pytorch library. On Yahoo! Answers, we train a BiLSTM sentence classifier which attains testing accuracy of 71%.

---

[2]We found that concurrent to our work, greedy with choosing the players with the highest restricted Banzhaf was used in Elkind et al. (2017).

Table 1: AUC of Robustness-$\overline{S_r}$ and Robustness-$S_r$ for various explanations on different datasets. The higher the better for Robustness-$\overline{S_r}$; the lower the better for Robustness-$S_r$.

| Datasets | Explanations | Grad | IG | EG | SHAP | LOO | BBMP | CFX | Random | Greedy-AS |
|---|---|---|---|---|---|---|---|---|---|---|
| MNIST | Robustness-$\overline{S_r}$ | 88.00 | 85.98 | 93.24 | 75.48 | 74.14 | 78.58 | 69.88 | 64.44 | **98.01** |
| | Robustness-$S_r$ | 91.72 | 91.97 | 91.05 | 101.49 | 104.38 | 176.61 | 102.81 | 193.75 | **82.81** |
| ImageNet | Robustness-$\overline{S_r}$ | 27.13 | 26.01 | 26.88 | 18.25 | 22.29 | 21.56 | 27.12 | 17.98 | **31.62** |
| | Robustness-$S_r$ | 45.53 | 46.28 | 48.82 | 60.02 | 58.46 | 158.01 | 46.10 | 56.11 | **43.97** |
| Yahoo!Answer | Robustness-$\overline{S_r}$ | 1.97 | 1.86 | 1.96 | 1.81 | 1.74 | - | 1.95 | 1.71 | **2.13** |
| | Robustness-$S_r$ | 2.91 | 3.14 | 2.99 | 3.34 | 4.04 | - | 2.96 | 7.64 | **2.41** |

**Setup.** In the experiments, we consider $p = 2$ for $\| \cdot \|_p$ in Eqn. 1 and Eqn. 2. We note that Eqn. 1 is defined for a single data example. Given $n$ multiple examples $\{x_i\}_{i=1}^n$ with their corresponding relevant sets provided by some explanation $\{S_i\}_{i=1}^n$, we compute the overall Robustness-$\overline{S_r}$ (-$S_r$) by taking the average: for instance, $\frac{1}{n}\sum_{i=1}^n g(f, x_i, \overline{S_i})$ for Robustness-$\overline{S_r}$. We then plot the evaluation curve as discussed in Section 3.2 and report the AUC for different explanation methods. For all quantitative results, we report the average over 100 random examples. For the baseline methods, we include vanilla gradient (Grad) (Shrikumar et al., 2017), integrated gradient (IG) (Sundararajan et al., 2017), and expected gradient (EG) (Erion et al., 2019; Sturmfels et al., 2020) from gradient-based approaches; leave-one-out (LOO) (Zeiler & Fergus, 2014; Li et al., 2016), SHAP (Lundberg & Lee, 2017) and black-box meaningful perturbation (BBMP) (only for image examples) (Fong & Vedaldi, 2017) from perturbation-based approaches (Ancona et al., 2018); counterfactual explanation (CFX) proposed by Wachter et al. (2017); Anchor (Ribeiro et al., 2018) for text examples; and a Random baseline that ranks feature importance randomly. Following common setup (Sundararajan et al., 2017; Ancona et al., 2018), we use zero as the reference value for all explanations that require baseline. We leave more implementation detail in Appendix C due to space limitation.

## 5.1 ROBUSTNESS ANALYSIS ON MODEL INTERPRETABILITY METHODS

Here we compare Greedy-AS and various existing explanation methods under the proposed evaluation criteria Robustness-$\overline{S_r}$ and Robustness-$S_r$. We list the results in Table 1, with the detailed plots in Appendix D. We as well compare Greedy-AS with its ablated variants in Appendix E.

**Comparisons between Different Explanations.** From Table 1, we observe that the proposed Greedy-AS consistently outperforms other explanation methods on both criteria (see Appendix G for the statistical significance). On one hand, this suggests that the proposed algorithm indeed successfully optimizes towards the criteria; on the other hand, this might indicate the proposed criteria do capture different characteristics of explanations which most of the current explanations do not possess. Another somewhat interesting finding from the table is that while Grad has generally been viewed as a baseline method, it nonetheless performs competitively on the proposed criteria. We conjecture the phenomenon results from the fact that Grad does not assume any reference value as opposed to other baselines such as LOO which sets the reference value as zero to mask out the inputs. Indeed, it might not be surprising that Greedy-AS achieves the best performances on the proposed criteria since it is explicitly designed for so. To more objectively evaluate the usefulness of the proposed explanation, we demonstrate different advantages of our method by comparing Greedy-AS to other explanations quantitatively on existing commonly adopted measurements, and qualitatively through visualization in the following subsections.

## 5.2 EVALUATING GREEDY-AS

**The Insertion and Deletion Metric.** To further justify the proposed explanation not only performs well on the very metric it optimizes, we evaluate our method on the suite of quantitative measurements mentioned above: the Insertion and Deletion criteria (Samek et al., 2016; Petsiuk et al., 2018; Sturmfels et al., 2020). Recall that evaluations on Insertion and Deletion score require specifying a reference value to represent feature missingness. Here, we first focus on the results when the reference values are randomly sampled from an uniform distribution, i.e., $\mathcal{U}(0, 1)$ for image inputs and random word vector for text inputs, and we shall discuss the impact on varying such reference value shortly. We plot the evaluation curves (in Appendix F) and report corresponding AUCs in Table 2. On these additional two criteria, we observe that Greedy-AS performs favorably against other explanations (see Appendix G for the statistical significance). The results further validate the benefits of the proposed criteria where optimizing Robustness-$\overline{S_r}$ (-$S_r$) has tight connection to optimizing the

Table 2: AUC of the Insertion and Deletion criteria for various explanations on different datasets. The higher the better for Insertion; the lower the better for Deletion.

| Datasets | Explanations | Grad | IG | EG | SHAP | LOO | BBMP | CFX | Random | Greedy-AS |
|---|---|---|---|---|---|---|---|---|---|---|
| MNIST | Insertion | 174.18 | 177.12 | 228.64 | 125.93 | 121.99 | 108.97 | 102.05 | 51.71 | **270.75** |
| | Deletion | 153.58 | 150.90 | 113.21 | 213.32 | 274.77 | 587.08 | 137.69 | 312.07 | **94.24** |
| ImageNet | Insertion | 86.16 | 109.94 | 150.81 | 28.06 | 63.90 | 135.98 | 97.33 | 31.73 | **183.66** |
| | Deletion | 276.78 | 256.51 | 244.88 | **143.27** | 290.10 | 615.13 | 281.12 | 314.82 | 219.52 |
| Yahoo!Answers | Insertion | 0.06 | 0.06 | 0.20 | 0.07 | 0.18 | - | 0.05 | 0.10 | **0.21** |
| | Deletion | 2.57 | 2.96 | 2.07 | 2.23 | 2.07 | - | 2.35 | 2.63 | **1.56** |

Insertion (Deletion) score. We note that on ImageNet, SHAP obtains a better performance under the Deletion criterion. We however suspect such performance comes from adversarial artifacts (features vulnerable to perturbations while not being semantically meaningful) since SHAP seems rather noisy on ImageNet (as shown in Figure 3), and the Deletion criterion has been observed to favor such artifacts in previous work (Dabkowski & Gal, 2017; Chang et al., 2019). We note that although Greedy-AS exploits regions that are most susceptible to adversarial attacks, such regions may still be meaningful as shown in our visualization result.

**Impact and Potential Bias of Reference Value.** From Table 2, one might wonder why explanations like IG and SHAP would suffer relatively low performance although some of these methods (e.g., SHAP) are intentionally designed for optimizing Insertion- and Deletion-like measurements. We anticipate that such inferior performances are due to the mismatch between the intrinsic reference values used in the explanations and the ones used in the evaluation (recall that we set the intrinsic reference value to zero for all explanation methods, but utilize random value for Insertion and Deletion). To validate the hypothesis, we evaluate all explanations on Insertion and Deletion criteria with different reference values (0, 0.25, 0.5, 0.75, 1), and list the results in Appendix H. We validate that zero-baseline SHAP and IG perform much stronger when the reference value used in Insertion / Deletion is closer to zero (matching the intrinsically-used baseline) and perform significantly worse when the reference value is set to 0.75, 1, or $\mathcal{U}(0, 1)$. On the other hand, we observe EG that does not rely on a single reference point (but instead averaging over multiple baselines) performs much more stably across different reference values. Finally, we see that Greedy-AS performs stably among the top across different reference values, which could be the merits of not assuming any baselines (but instead consider the worst-case perturbation). These empirical results reveal potential risk of evaluations (and indeed explanations) that could largely be affected by the change of baseline values.

**Sanity Check and Sensitivity Analysis.** In Appendix I, we conduct experiments to verify that Greedy-AS could indeed pass the sanity check proposed by Adebayo et al. (2018). We as well conduct sensitivity analysis (Yeh et al., 2019) on Greedy-AS in Appendix J.

## 5.3 QUALITATIVE RESULTS

**Image Classification.** To complement the quantitative measurements, we show several visualization results on MNIST and ImageNet in Figure 2 and Figure 3. More examples could be found in Appendix K and L. On MNIST, we observe that existing explanations tend to highlight mainly on the white pixels in the digits; among which SHAP and LOO show less noisy explanations comparing to Grad and IG. On the other hand, the proposed Greedy-AS focuses on both the "crucial positive" (important white pixels) as well as the "pertinent negative" (important black regions) that together support the prediction. For example, in the first row, a 7 might have been predicted as a 4 or 0 if the pixels highlighted by Greedy-AS are set to white. Similarly, a 1 may be turned to a 4 or a 7 given additional white pixels to its left, and a 9 may become a 7 if deleted the lower circular part of its head. From the results, we see that Greedy-AS focuses on *"the region where perturbation on its current value will lead to easier prediction change"*, which includes both the crucial positive and pertinent negative pixels. Such capability of Greedy-AS is also validated by its superior performance on the proposed robustness criteria, on which methods like LOO that highlights only the white strokes of digits show relatively low performance. The capability of capturing pertinent negative features has also been observed in explanations proposed in some recent work Dhurandhar et al. (2018); Bach et al. (2015); Oramas et al. (2019), and we further provide more detailed discussions and comparisons to these methods in Appendix M. From the visualized ImageNet examples shown in Figure 3, we observe that our method provides more compact explanations that focus mainly on the actual objects being classified. For instance, in the first image, our method focuses more on the face of the Maltese while others tend to have noisier results; in the last image, our method focuses on one of the Japanese Spaniel whereas others highlight both the dogs and some noisy regions.

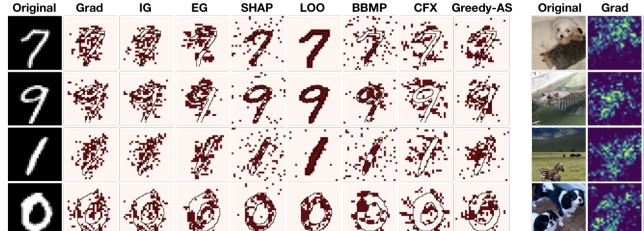 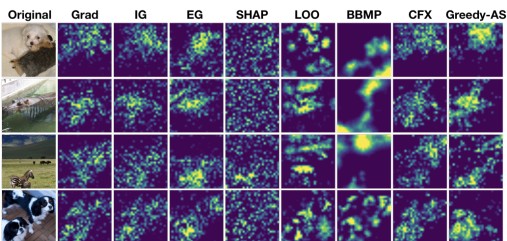

Figure 2: Visualization on top 20 percent relevant features provided by different explanations on MNIST. We see Greedy-AS highlights both crucial positive and pertinent negative features supporting the prediction.

Figure 3: Visualization of different explanations on ImageNet, where the predicted class for each input is "Maltese", "hippopotamus", "zebra", and "Japanese Spaniel". Greedy-AS focuses more compactly on objects.

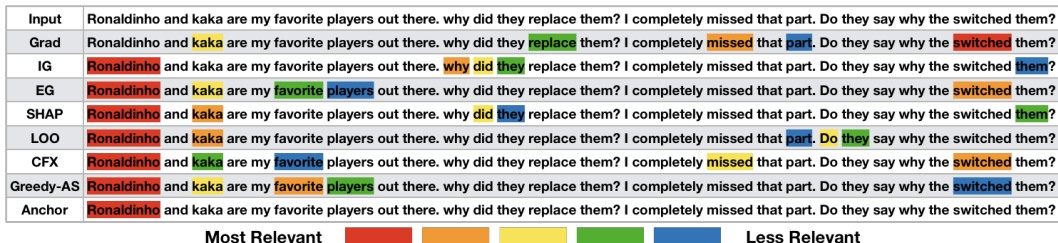

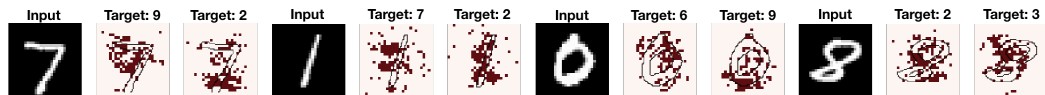

Figure 4: Explanations on a text classification model which correctly predicts the label "sport". Unlike most other methods, the top-5 relevant keywords highlighted by Greedy-AS are all related to the concept "sport".

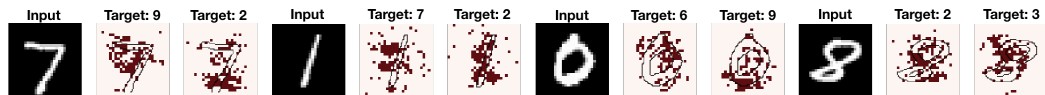

Figure 5: Visualization of targeted explanation. For each input, we highlight relevant regions explaining why the input is not predicted as the target class. We see the explanation changes in a semantically meaningful way as the target class changes.

**Text Classification.** Here we demonstrate our explanation method on a text classification model that classifies a given sentence into one of the ten classes (Society, Science, ..., Health). We showcase an example in Figure 4 (see Appendix O for more examples). We see that the top-5 keywords highlighted by Greedy-AS are all relevant to the label "sport", and Greedy-AS is less likely to select stop words as compared to other methods. Additionally, we conduct an user study where we observe that Greedy-AS generates explanation that matches user intuition the most, and we report detailed setup of the user study in Appendix P.

**Targeted Explanation Analysis.** In section 3.2, we discussed about the possibility of using targeted adversarial perturbation to answer the question of *"why the input is predicted as A but not B"*. In Figure 5, for each input digit, we provide targeted explanation towards two different target classes. Interestingly, as the target class changes, the generated explanation varies in an interpretatble way. For example, in the first image, we explain why the input digit 7 is not classified as a 9 (middle column) or a 2 (rightmost column). The resulting explanation against 9 highlights the upper-left part of the 7. Semantically, this region is indeed pertinent to the classification between 7 and 9, since turning on the highlighted pixel values in the region (currently black in the original image) will then make the 7 resemble a 9. However, the targeted explanation against 2 highlights a very different but also meaningful region, which is the lower-right part of the 7; since adding a horizontal stroke on the area would turn a 7 into a 2.

# 6 CONCLUSION

In this paper, we establish the link between a set of features to a prediction with a new evaluation criteria, robustness analysis, which measures the minimum tolerance of adversarial perturbation. Furthermore, we develop a new explanation method to find important set of features to optimize this new criterion. Experimental results demonstrate that the proposed new explanations are indeed capturing significant feature sets across multiple domains.

ACKNOWLEDGEMENTS

This work is partially supported by NSF IIS-1901527, IIS-2008173 and IIS-2048280. C.Y. and P.R. acknowledge the support of DARPA via FA87501720152, HR00112020006.

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

## A    Bias for Reference-Value Methods

In Appendix A.1, we show that many explanations are biased to reference value (IG, SHAP, LRP, DeepLift). For example, if the reference value is a zero vector (which means black pixels in image), then any black pixel will get a zero attribution value no matter if the object of interest is actually back. We note that the Expected Gradient (Erion et al., 2019) is not biased to reference value by this theoretic definition since the baseline is actually a distribution. However, for feature values that are close to the distribution of , the attribution score will be lower (but not 0 as our theoretic definition), when feature values are far from the distribution of , the the attribution score will be larger, which still has some biased involve. We leave further investigation of the problem to future work to use more advanced analysis to quantify such a bias for explanations when the baseline follows a distribution.

In Appendix A.2, we added theoretical analysis that when a feature is equal to the replaced reference value, no matter how important it actually is, it will not contribute to the Deletion score nor the Insertion score. However, the lower the deletion score is better, and the higher the insertion score is better, and choosing an important feature that corresponds to the reference value will inevitably not improve the Deletion score and Insertion score. The reference values such as blurring out or adding noise may still make the original features unchanged (such as when the main part of the image is already blurred, blurring the image will not change the image, and thus the main part of the image is biased to blurred baseline).

### A.1    Bias for Reference-Value Based Explanations

**Theorem A.1** *For any reference-value based explanation with the form $\phi(f, x, x') = (x - x') \otimes K(f, x, x')$ for any meta-function K, we know that if $x_i = x'_i$, $\phi_i(f, x) = 0$. That is, when ever a feature value coincides with the baseline value $x'$, it will get 0 attribution even if it is the main feature contributing to the prediction. We call such explanations biased to the reference value $x'$.*

**Corollary A.1** *IG, LRP, DeepLift are biased to the reference value $x'$.*

Following the definition in (Ancona et al., 2018), let $\frac{\partial^g x_c}{\partial x_i} = \sum_{p \in P_{ic}} \prod w_p \prod g(z)_p$ where $P_{ic}$ is the set of all paths that connect an unit $i$ to unit $c$ in a deep neural network, $w_p$ are the weights existing in path $p$, $z$ be the value of an unit before nonlinear activation, and $g()$ be any generic function. Note that if $g() = f'()$, the definition corresponds to the standard partial derivative of unit $c$ to unit $i$.

$$\phi_i^{IG}(f, x) = (x_i - x'_i) \cdot K(f, x, x'), \text{ where } K(f, x, x') = \int_{\alpha=0}^{1} \frac{\partial f(x' + \alpha(x - x'))}{\partial x_i} d\alpha \qquad (7)$$

$$\phi_i^{LRP}(f, x) = (x_i - x'_i) \cdot K(f, x, x'), \text{ where } x' = 0 \text{ and } K(f, x, x') = \frac{\partial^g f(x)}{\partial x_i} \text{ with } g = \frac{f(z)}{z} \qquad (8)$$

$$\phi_i^{DeepLift}(f, x) = (x_i - x'_i) \cdot K(f, x, x'), \text{ where } K(f, x, x') = \frac{\partial^g f(x)}{\partial x_i} \text{ with } g = \frac{f(z) - f(z')}{z - z'} \qquad (9)$$

For example, if the reference value $x'$ is a zero vector (which means black pixels in image), then any black pixel will get a zero attribution value no matter if the object of interest is actually back.

**Theorem A.2** *For any average reference-value based explanation with the form $\phi(f, x, x') = \mathbb{E}_{x' \sim p_b}[(x - x') \otimes K(f, x)]$ for some meta function K, if $x_i = \mathbb{E}_{x' \sim p_b}[x'_i]$, $\phi_i(f, x) = 0$. That is, they are biased to the reference value $\mathbb{E}_{x' \sim p_b}[x']$.*

**Corollary A.2** *Averaging LRP over multiple baselines are bias to the reference value $\mathbb{E}_{x' \sim p_b}[x']$.*

$$\phi_i^{avgLRP}(f, x, x') = \mathbb{E}_{x' \sim p_b}[(x_i - x'_i) \cdot K(f, x)], \qquad (10)$$

$$\text{where } K(f, x) = \frac{\partial^g f(x)}{\partial x_i} \text{ with } g = \frac{f(z)}{z}$$

**Theorem A.3** *The baseline Shapley value (Sundararajan & Najmi, 2019) is biased to reference value $x'$.*

$$\phi_i^{BShap}(f, x, x') = \sum_{S \subseteq N \setminus i} \frac{|S|! \cdot (|N| - |S| - 1)!}{|N|!} (v(S \cup i) - v(S)) \tag{11}$$

where $v(s) = f(x_S; x'_{N \setminus S})$ and $N$ is the set of all features.

These above analysis shows that more explanations are also biased to reference value. We note that by Expected Gradient (Erion et al., 2019) is not biased to reference value by this theoretic definition since the baseline $x'$ is actually a distribution. However, for feature values $x_i$ that are close to the distribution of $x'_i$, the attribution score will be lower (but not 0 as our theoretic definition), when feature values $x_i$ are far from the distribution of $x'_i$, the the attribution score will be larger, which still shows some level of bias. We leave this to future work to use more advanced analysis techniques to quantify such a biased for explanations when the baseline follows a distribution.

## A.2 BIAS FOR REFERENCE-VALUE BASED EVALUATIONS

**Definition A.1** *(Informal) Given a data point $x$, a reference point $x'$, the deletion score of a model $f$ given a set of important features $S$ (and denote the complement of $S$ as $\bar{S}$), which we abbreviate as DEL$(x, x', S)$, is defined as $f(x'_S; x_{\bar{S}})$. Here, $x'$ can be a fixed value or a random value, or even a blurred component (by a slight abuse of notations). Similarly, we define the insertion score for a data point $x$, a reference point $x'$, important set of features $S$, and a model $f$ as INSR$(x, x', S) = f(x_S; x'_{\bar{S}})$.*

Note that the lower the DEL score is better, and the higher the INSR score is better.

**Theorem A.4** *If $x_i = x'_i$, then DEL$(x, x', S) = $ DEL$(x, x', S \cup i)$ for $i \not\subseteq S$. Similarly, if $x_i = x'_i$, INSR$(x, x', S) = $ INSR$(x, x', S \cup i)$ for $i \not\subseteq S$*

The implication of Thm. A.4 is that if a feature happens to correspond to the reference value, no matter how important it actually is, it will not contribute to the DEL score nor the INSR score. However, the lower the DEL score is better, and the higher the INSR score is better, and choosing an important feature that corresponds to the reference value will inevitably not improve the DEL score and INSR score. Thus, DEL score and INSR score will deem feature with the same value as the reference value as non-important even if the features are actually crucial.

# B MORE ON ESTIMATING ADVERSARIAL ROBUSTNESS

Our work relies on measuring the adversarial perturbation norm on a subset of features, which can be seen as a constrained adversarial robustness problem. Adversarial robustness has been extensively studied in the past few years. The adversarial robustness of a machine learning model on a given sample can be defined as the shortest distance from the sample to the decision boundary, which corresponds to our definition in Eqn. 1. Algorithms have been proposed for finding adversarial examples (feasible solutions of Eqn. 1), including (Goodfellow et al., 2014; Carlini & Wagner, 2017; Madry et al., 2017). However, those algorithms only work for neural networks, while for other models such as tree based models or nearest neighbor classifiers, adversarial examples can be found by decision based attacks (Brendel et al., 2017; Cheng et al., 2018; Chen et al., 2019). Therefore the proposed framework can also be used in other decision based classifiers. On the other hand, several works aim to solve the neural network verification problem, which is equivalent to finding a lower bound of Eqn. 1. Examples include (Singh et al., 2018; Wong & Kolter, 2018; Zhang et al., 2018). In principal, our work can also apply these verification methods for getting an approximate solution of Eqn. 1, but in practice they are very slow to run and often gives loose lower bounds on regular trained networks.

## C  IMPLEMENTATION DETAIL

**Underlying Models Used.**   In the experiments conducted on MNIST dataset, we train a convolutional neural network (CNN) with 99% testing accuracy. The training and testing split used in the experiments are the default split as provided by the original dataset. For each instance, the inputs are loaded into the range of $[0, 1]$. On ImageNet dataset, we deploy a pre-trained ResNet model obtained from the Pytorch library.[3] For each image, the inputs are first loaded into the range of $[0, 1]$ and normalized using mean $= [0.485, 0.456, 0.406]$ and standard deviation $= [0.229, 0.224, 0.225]$. On Yahoo! Answers dataset, we train a BiLSTM sentence classifier which attains testing accuracy of 71%. For the model configuration and training hyperparameters, we strictly follow the official GLUE repository [4]. Specifically, we first tokenize the dataset by word-level tokenizer; and then we use the pretrained GloVe word embedding to initialize the LSTM embedding layer where the hidden size is 300.

**Method Implementation Details and Hyperparameters.**   For the proposed Greedy and Greedy-AS, at each greedy iteration, we include the top-5% features with highest scores into the relevant set to further speed up the selection process. As discussed in Section 3.2, we use PGD attack with binary search to approximately compute the robustness value in Eqn. 1. In the experiments, we set the PGD attack step size to be 1.0 and number of steps to be 100. The hyperparameters are chosen such that the PGD attack could most efficiently provide the tightest upper bound on the true robustness value. As mentioned in Section 4.2, we solve Eqn. 6 by subsampling from all possible subsets of $\overline{S_r^t}$. Specifically, we compute the coefficients $\boldsymbol{w}$ with respect to 5000 sampled subsets when learning the regression.

For CFX, since the original objective in (Wachter et al., 2017) has no control over the number of features that could differ from the input, we add $L_0$-norm constraint to enforce sparsity in the difference and enable selection of top-$K$ most relevant features.

For text classification models, in the experiments, we evaluate the top-5 keywords selected by different explanations with the proposed Robustness-$\overline{S_r}$ (-$S_r$) measurements and the existing Insertion and Deletion scores with different baseline values. In the experiments, we represent a length-$n$ sentence by $n$ embedding vectors where each embedding vector is itself $d$-dimensional. To calculate the Robustness-$\overline{S_r}$ of an input sentence where the relevant set contains a set of indices of the top-$k$ keywords, we restrict the embedding vectors of the top-$k$ words to be fixed and only allow perturbations on the remaining $(n - k) \cdot d$ dimensions. Similarly for Robustness-$S_r$, we only allow perturbations on the $d \cdot k$ dimensions corresponding to the embedding vectors of the top-$k$ keywords. For Insertion and Deletion score, we follow the same to replace the embedding vectors of the masked words by some selected reference value.

**Evaluation Curves.**   To plot the evaluation curves of a given explanation, we measure the explanation's Robustness-$\overline{S_r}$ and Robustness-$S_r$ at different sizes of relevant set $|S_r|$. Specifically, we measure the explanation's performance when $|S_r|$ equals to 5%, 10%, ..., 45% of the total number of features. Similarly for the Insertion and Deletion criteria, we progressively evaluate the explanation's performance when 5%, 10%, ..., 45% of the features are inserted (or removed) from the inputs. After the evaluation curves are plotted, we then calculate the area under curves to summarize the explanation's overall performance under each criterion.

**Hardware and Code.**   All the experiments were performed on Intel(R) Xeon(R) CPU E5-2630 v4 @ 2.20GHz and NVIDIA GeForce GTX 1080 Ti GPU. We release our source code along with an example Jupyter notebook for more information.

---

[3]https://github.com/pytorch/pytorch
[4]https://github.com/nyu-mll/GLUE-baselines

# D    EVALUATION CURVES UNDER ROBUSTNESS-$\overline{S_r}$ AND ROBUSTNESS-$S_r$

We show the evaluation curves of different methods under the proposed robustness criteria on MNIST, ImangeNet, and Yahoo!Answers in Figure 6, Figure 7, and Figure 8 respectively.

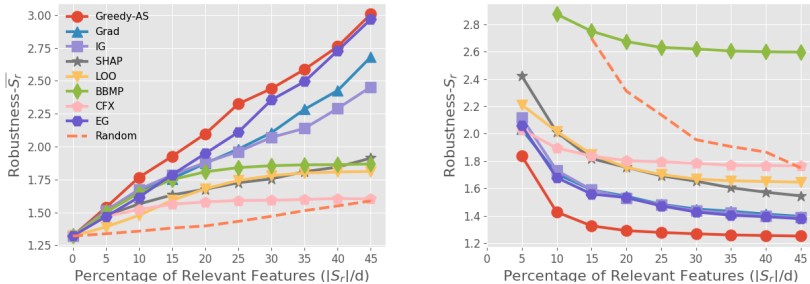

Figure 6: Greedy-AS and other existing methods on MNIST.

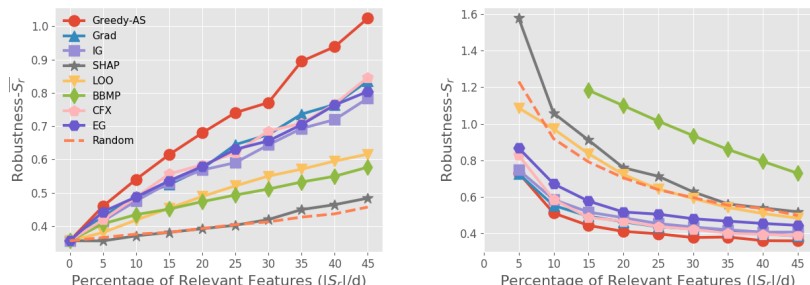

Figure 7: Greedy-AS and other existing methods on ImageNet.

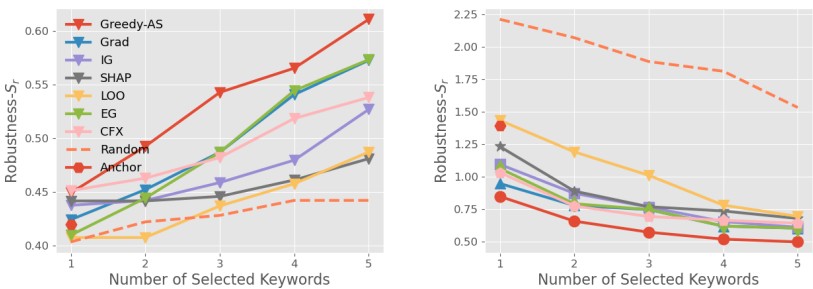

Figure 8: Greedy-AS and other existing methods on Yahoo!Answers.

Table 3: AUC of Robustness-$\overline{S_r}$ and Robustness-$S_r$ for Greedy-AS and its variants. The higher the better for Robustness-$\overline{S_r}$; the lower the better for Robustness-$S_r$.

| Datasets | Explanations | Greedy-AS | Greedy | One-Step Banzhaf |
|---|---|---|---|---|
| MNIST | Robustness-$\overline{S_r}$ | **98.01** | 83.57 | 86.37 |
| | Robustness-$S_r$ | **82.81** | 171.56 | 83.59 |
| ImageNet | Robustness-$\overline{S_r}$ | **31.62** | 21.16 | 24.54 |
| | Robustness-$S_r$ | **43.97** | 58.45 | 47.07 |

# E  ABLATION STUDY ON GREEDY-AS

As discussed in Section 4.2, Greedy-AS could be seen as a combination of the original greedy procedure with the approximated contribution of each feature computed by a regression. Here, we examine the importance of both components by comparing the Greedy-AS method to two baselines, where one selects important features based only on the pure Greedy method (Section 4.1) and the other utilizes only a single step of regression without the iterative greedy procedure. As the latter essentially corresponds to the Banzhaf value, we term this method as *One-Step Banzhaf*. As shown in Table 3, the pure Greedy method suffers degraded performances comparing to Greedy-AS under both criteria. The inferior performance could be explained by the ignorance of feature correlations which ultimately results in the introduction of noise (see Figure 9). In addition, we also see that Greedy-AS performs better than One-Step Banzhaf. This could result from the fact that One-Step Banzhaf considers the feature interactions among all features with equal probability. However, in our objective, we only care about those interactions with the most important features. By iteratively selecting the features with highest Banzhaf value in Greedy-AS, we give more weight on the interactions among the most important features through iterations, and as a result lead to better performance.

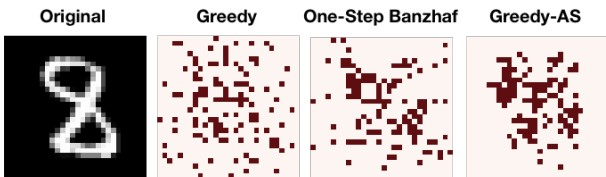

Figure 9: Visualization on our proposed methods. The top features selected by Greedy-AS are less noisy.

# F EVALUATION CURVES UNDER INSERTION AND DELETION

We show the evaluation curves of different methods under the Insertion and Deletion criteria on MNIST and ImageNet in Figure 10, Figure 11, and Figure 12 respectively.

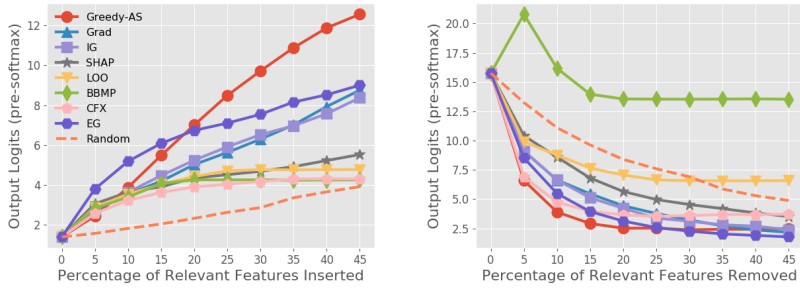

Figure 10: Greedy-AS and other existing methods on MNIST.

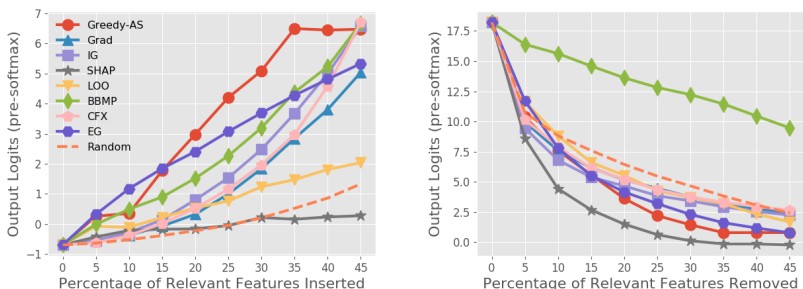

Figure 11: Greedy-AS and other existing methods on ImageNet.

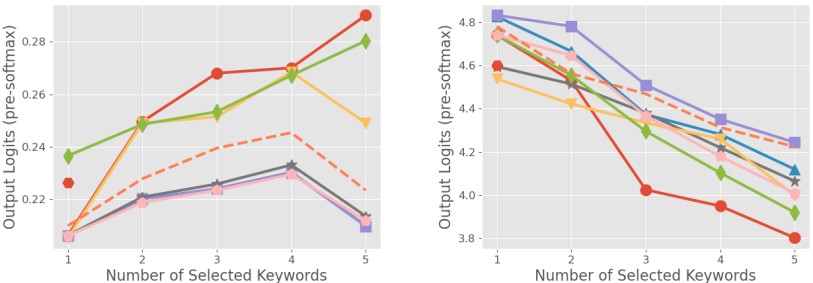

Figure 12: Greedy-AS and other existing methods on Yahoo!Answers.

# G STUDENT'S $t$-TEST ON THE PERFORMANCE IMPROVEMENT OF GREEDY-AS OVER EXISTING METHODS

We conduct a set of Student's $t$-tests to further verify that the outperformances of the proposed Greedy-AS over existing methods are indeed statistically significant. We compare the AUCs of Greedy-AS and other methods across different criteria. In Table 4, we show the $t$-test results on the AUCs of Robustness-$\overline{S_r}$ and Robustness-$S_r$, as well as the results on the AUCs of Insertion and Deletion criteria. In the tables, *win* (*loss*) indicates that the performance improvement (drop) of Greedy-AS to the baselines is statistically significant; otherwise, we use *draw* to indicate insignificant difference between the performances. From the results, we positively verify the statistical significance of the performance improvements of our proposed method over existing explanations on various criteria.

Table 4: The proposed Greedy-AS versus other explanations under various criteria with Student's $t$-test at 95% confidence level.

| Datasets | Criteria | Greedy-AS vs. | | | | | |
| --- | --- | --- | --- | --- | --- | --- | --- |
| | | Grad | IG | SHAP | LOO | BBMP | CFX |
| MNIST | Robustness-$\overline{S_r}$ | win | win | win | win | win | win |
| | Robustness-$S_r$ | win | win | win | win | win | win |
| | Insertion | win | win | win | win | win | win |
| | Deletion | win | win | win | win | win | win |
| ImageNet | Robustness-$\overline{S_r}$ | win | win | win | win | win | win |
| | Robustness-$S_r$ | draw | win | win | win | win | draw |
| | Insertion | win | win | win | win | win | win |
| | Deletion | draw | draw | loss | win | win | draw |

# H    ANALYSIS ON THE EFFECT OF REFERENCE VALUE

We provide in Table 5 and Table 6 an empirical analysis on how different reference values would affect the evaluation results of different explanation methods. The corresponding evaluation curves are shown in Figure 14, Figure 15, Figure 16, and Figure 17. We as well summarize the performances of different explanations across different reference values in Figure 13. We see that Greedy-AS performs most stably among the top across different reference values, while methods like IG, SHAP, and LOO suffer performance degradation when the reference value (used in Insertion/Deletion) moves away from zero, the intrinsic reference value these methods rely on.

Table 5: AUC of the Insertion and Deletion criteria with different reference values for various explanations on MNIST. The higher the better for Insertion; the lower the better for Deletion.

| Reference Values | Explanations | Grad | IG | SHAP | LOO | BBMP | CFX | EG | Random | Greedy-AS |
|---|---|---|---|---|---|---|---|---|---|---|
| Reference value = rand | Insertion | 174.18 | 177.12 | 125.93 | 121.99 | 108.97 | 102.05 | 228.64 | 51.71 | **270.75** |
| rand ~ Unifrom(0, 1) | Deletion | 153.58 | 150.90 | 213.32 | 274.77 | 587.08 | 137.69 | 113.21 | 312.07 | **94.24** |
| Reference value = 0 | Insertion | 393.25 | **802.57** | 656.55 | 706.63 | 212.12 | 81.59 | 450.37 | 120.65 | 502.32 |
| | Deletion | 214.81 | 134.64 | **66.36** | 119.91 | 362.95 | 840.78 | 138.57 | 453.01 | 252.44 |
| Reference value = 0.25 | Insertion | 293.92 | 424.45 | 412.87 | 453.29 | 142.00 | 77.33 | 341.05 | 78.73 | **455.74** |
| | Deletion | 162.24 | 133.06 | 115.88 | 204.75 | 442.23 | 354.05 | 161.28 | 426.21 | **90.89** |
| Reference value = 0.5 | Insertion | 218.18 | 317.55 | 136.51 | 189.87 | 42.76 | 86.80 | 382.31 | 49.15 | **479.92** |
| | Deletion | 209.83 | 258.50 | 271.11 | 424.66 | 549.11 | **90.50** | 195.47 | 392.69 | 155.59 |
| Reference value = 0.75 | Insertion | 220.90 | 204.92 | 79.28 | 125.73 | 21.97 | 163.46 | 285.66 | 67.35 | **325.47** |
| | Deletion | 305.17 | 305.38 | 481.67 | 712.46 | 708.93 | **89.38** | 250.57 | 451.18 | 176.85 |
| Reference value = 1 | Insertion | 234.61 | 206.54 | 89.83 | 128.71 | 25.31 | 229.02 | 276.48 | 81.44 | **313.66** |
| | Deletion | 364.01 | 372.18 | 647.68 | 956.88 | 823.90 | **101.31** | 310.50 | 495.91 | 223.99 |

Table 6: AUC of the Insertion and Deletion criteria with different reference values for various explanations on ImageNet. The higher the better for Insertion; the lower the better for Deletion.

| Reference Values | Explanations | Grad | IG | SHAP | LOO | BBMP | CFX | EG | Random | Greedy-AS |
|---|---|---|---|---|---|---|---|---|---|---|
| Reference value = rand | Insertion | 86.16 | 109.94 | 28.06 | 63.90 | 135.98 | 97.33 | 150.81 | 31.73 | **183.66** |
| rand ~ Uniform(0, 1) | Deletion | 276.78 | 256.51 | **143.27** | 290.10 | 615.13 | 281.12 | 244.88 | 314.82 | 219.52 |
| Reference value = 0 | Insertion | 125.32 | **214.85** | 85.57 | 61.37 | 138.00 | 135.31 | 183.47 | 69.57 | 182.61 |
| | Deletion | 313.04 | 260.75 | **181.11** | 262.78 | 665.49 | 333.98 | 288.52 | 312.66 | 234.34 |
| Reference value = 0.25 | Insertion | 291.06 | **329.24** | 39.74 | 163.81 | 243.92 | 287.58 | 245.46 | 103.63 | 293.30 |
| | Deletion | 408.95 | 395.37 | 328.50 | 365.03 | 686.75 | 421.56 | 433.78 | 516.56 | **322.50** |
| Reference value = 0.5 | Insertion | 301.78 | 343.77 | 45.77 | 178.18 | 250.25 | 270.03 | **348.66** | 129.65 | 305.87 |
| | Deletion | 431.17 | 412.70 | 345.84 | 388.64 | 689.08 | 421.89 | 420.31 | 546.30 | **321.11** |
| Reference value = 0.75 | Insertion | 222.63 | 217.26 | 29.39 | 131.92 | 243.44 | 215.11 | 248.74 | 87.38 | **286.40** |
| | Deletion | 364.28 | 340.24 | 279.51 | 325.71 | 701.97 | 375.87 | 329.11 | 402.97 | **262.36** |
| Reference value = 1 | Insertion | 109.89 | 136.10 | 55.29 | 89.26 | 135.06 | 109.44 | 170.96 | 59.23 | **189.96** |
| | Deletion | 245.57 | 218.09 | 207.57 | 234.28 | 658.54 | 329.20 | 203.84 | 217.39 | **191.96** |

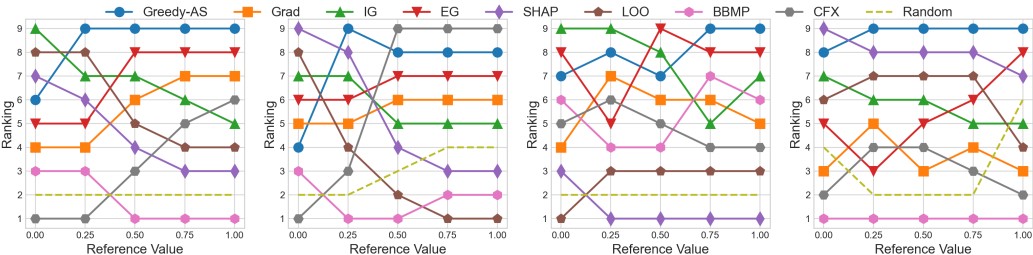

Figure 13: Ranking among different explanations. From left to right: MNIST Insertion/Deletion; ImageNet Insertion/Deletion. Rankings correspond to the relative rank among all methods, i.e. 1 - 9. We see performances of zero-baseline explanations, e.g. IG/SHAP/LOO, degrade as the reference value moves away from zero.

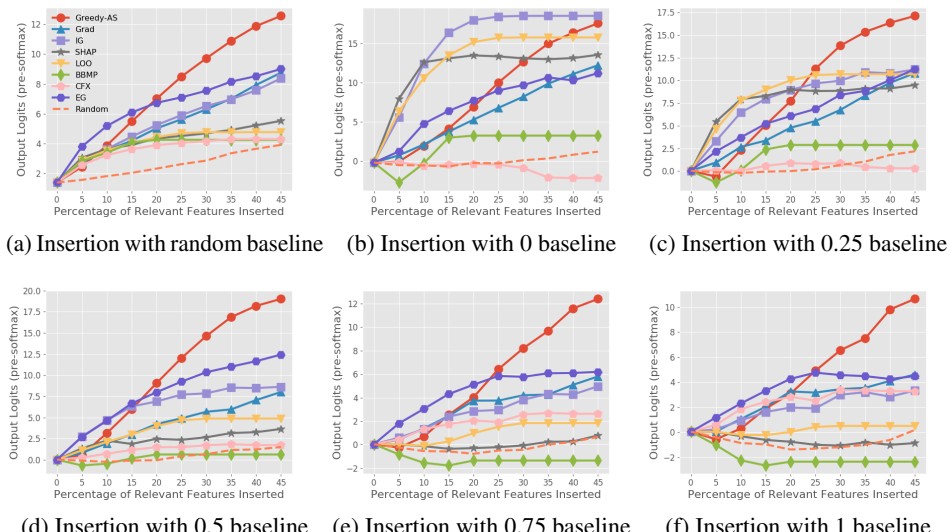

(a) Insertion with random baseline    (b) Insertion with 0 baseline    (c) Insertion with 0.25 baseline

(d) Insertion with 0.5 baseline    (e) Insertion with 0.75 baseline    (f) Insertion with 1 baseline

Figure 14: Evaluation curves of Insertion criteria on MNIST with different reference values.

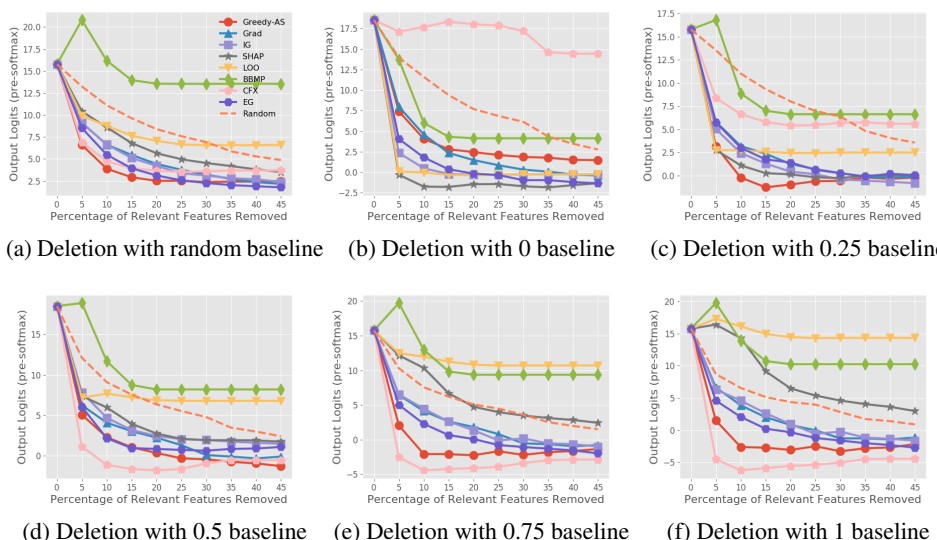

(a) Deletion with random baseline    (b) Deletion with 0 baseline    (c) Deletion with 0.25 baseline

(d) Deletion with 0.5 baseline    (e) Deletion with 0.75 baseline    (f) Deletion with 1 baseline

Figure 15: Evaluation curves of Deletion criteria on MNIST with different reference values.

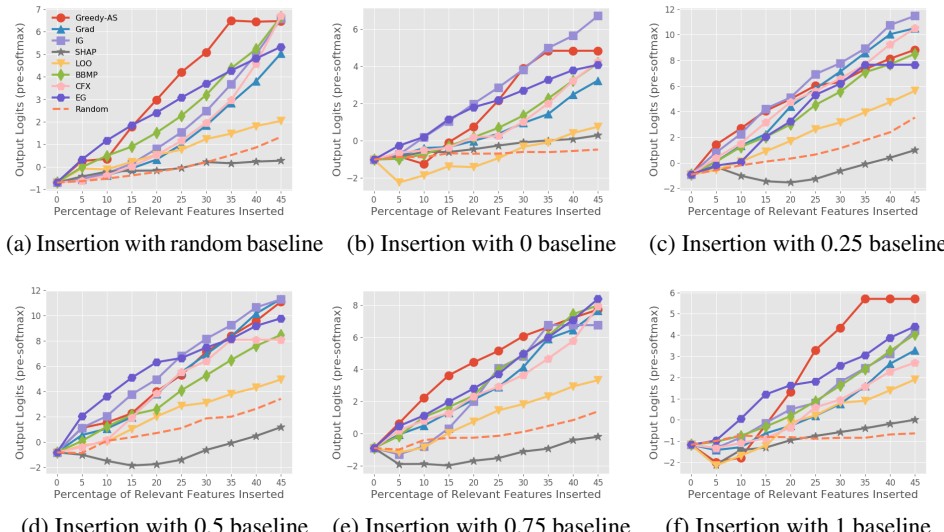

Figure 16: Evaluation curves of Insertion criteria on ImageNet with different reference values.

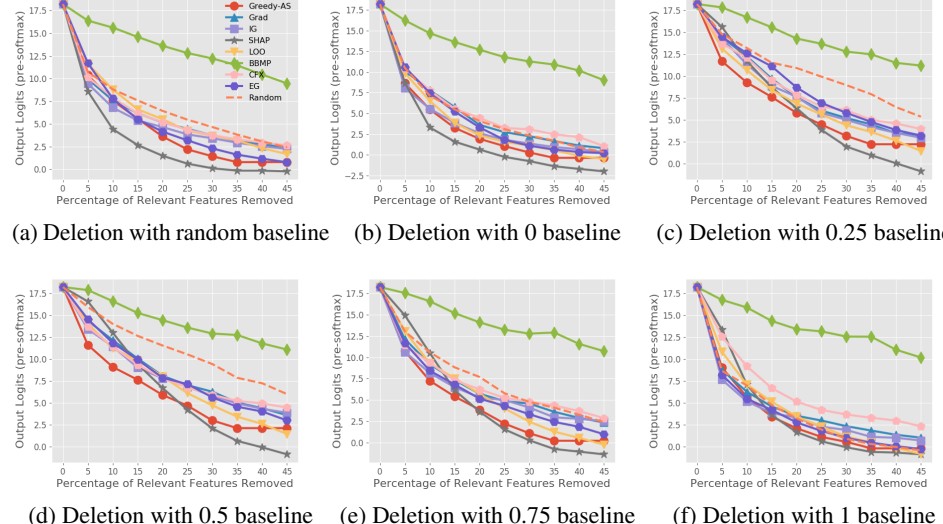

Figure 17: Evaluation curves of Deletion criteria on ImageNet with different reference values.

## I  SANITY CHECK

Recent literature has pointed out that an appropriate explanation should be related to the model being explained (Adebayo et al., 2018). To ensure that our proposed explanation does indeed reflect the model behavior, we conduct the sanity check proposed by Adebayo et al. (2018) to check if our explanations are adequately different when the model parameters are randomly re-initialized. In the experiment, we randomly re-initialize the last fully-connected layer of the neural network model. We then compute the rank correlation between explanation computed w.r.t. the original model and that w.r.t. the randomized model. From Table 7, we observe that Greedy-AS has a much lower rank correlation comparing to Grad, IG, and LOO, suggesting that Greedy-AS is indeed sensitive to model parameter change and is able to pass the sanity check.

Table 7: Rank correlation between explanations with respect to original and randomized model.

|       | Grad | IG   | SHAP | LOO  | BBMP | CFX  | Greedy-AS |
|-------|------|------|------|------|------|------|-----------|
| Corr. | 0.30 | 0.30 | 0.11 | 0.49 | 0.17 | 0.26 | 0.18      |

## J  EXPLANATION SENSITIVITY ANALYSIS

In a recent study, Yeh et al. (2019) proposed a sensitivity measurement to evaluate how large an explanation would change when small perturbation is applied to the original input. The measurement could be seen as the lower bound of the local Lipschitz (Alvarez Melis & Jaakkola, 2018) of an explanation functional, and could be calculated efficiently by sampling. Following Yeh et al. (2019), we define a similar sensitivity measurement for set explanation as follows.

**Definition J.1** *Given a black-box model to be explained $f$, an input $x$, an explanation functional $\Phi(f, x)$ that returns a set of relevant features, and a radius $r$ defining a local neighborhood around $x$, we define the sensitivity for an explanation as:*

$$SENS(\Phi, f, x, r) = 1 - \max_{\|y-x\| \leq r} \frac{|\Phi(f, y) \cap \Phi(f, x)|}{|\Phi(f, x)|}. \tag{12}$$

By Eqn. 12, we see that the sensitivity of an explanation equals 0 when the relevant set returned by the explanation remains the same in all local neighborhood of radius $r$ centered around the original input $x$. On the other hand, the sensitivity equals 1 when there exists some $y$ within the local neighborhood of $x$ where the relevant set returned by $\Phi(f, y)$ is totally disjoint from $\Phi(f, x)$. We shall however note that the sensitivity alone should not serve as the sole measurement for an explanation, since a meaningless constant explanation could always achieve 0 sensitivity. We show in Table 8 a sensitivity analysis of different explanations on MNIST dataset where we consider $\Phi(f, x)$ to be the top-20% relevant features selected by an explanation method, and set $r = 0.1$ in the experiments.

From Table 8, we see that Expected Gradient has the least sensitivity since EG could be seen as a variant of the smoothing technique (Smilkov et al., 2017) (discussed in (Sturmfels et al., 2020)), which is shown to be an effective approach to reduce explanation sensitivity (Yeh et al., 2019). While the proposed Greedy-AS has relatively large sensitivity, we believe such result might be highly correlated to the underlying sensitivity of the model $f$ itself (since Greedy-AS aims to faithfully reflect the decision boundary of the model around an input). Thus, one possible way to reduce the sensitivity of Greedy-AS is to consider an adversarially more robust underlying model whose decision boundary is less sensitive to perturbations. Another possible way is to consider smoothing the explanation provided by Greedy-AS. We leave further improvement on the sensitivity of Greedy-AS as a future direction.

Table 8: Sensitivity of Different Explanations.

| Explanation | Grad | IG   | LOO  | BBMP | SHAP | CFX  | EG   | Random | Greedy-AS |
|-------------|------|------|------|------|------|------|------|--------|-----------|
| Sensitivity | 0.45 | 0.38 | 0.31 | 0.38 | 0.52 | 0.57 | 0.20 | 0.85   | 0.61      |

## K    VISUALIZATION ON MNIST

We provide more examples on visualized explanations on MNIST in Figure 18.

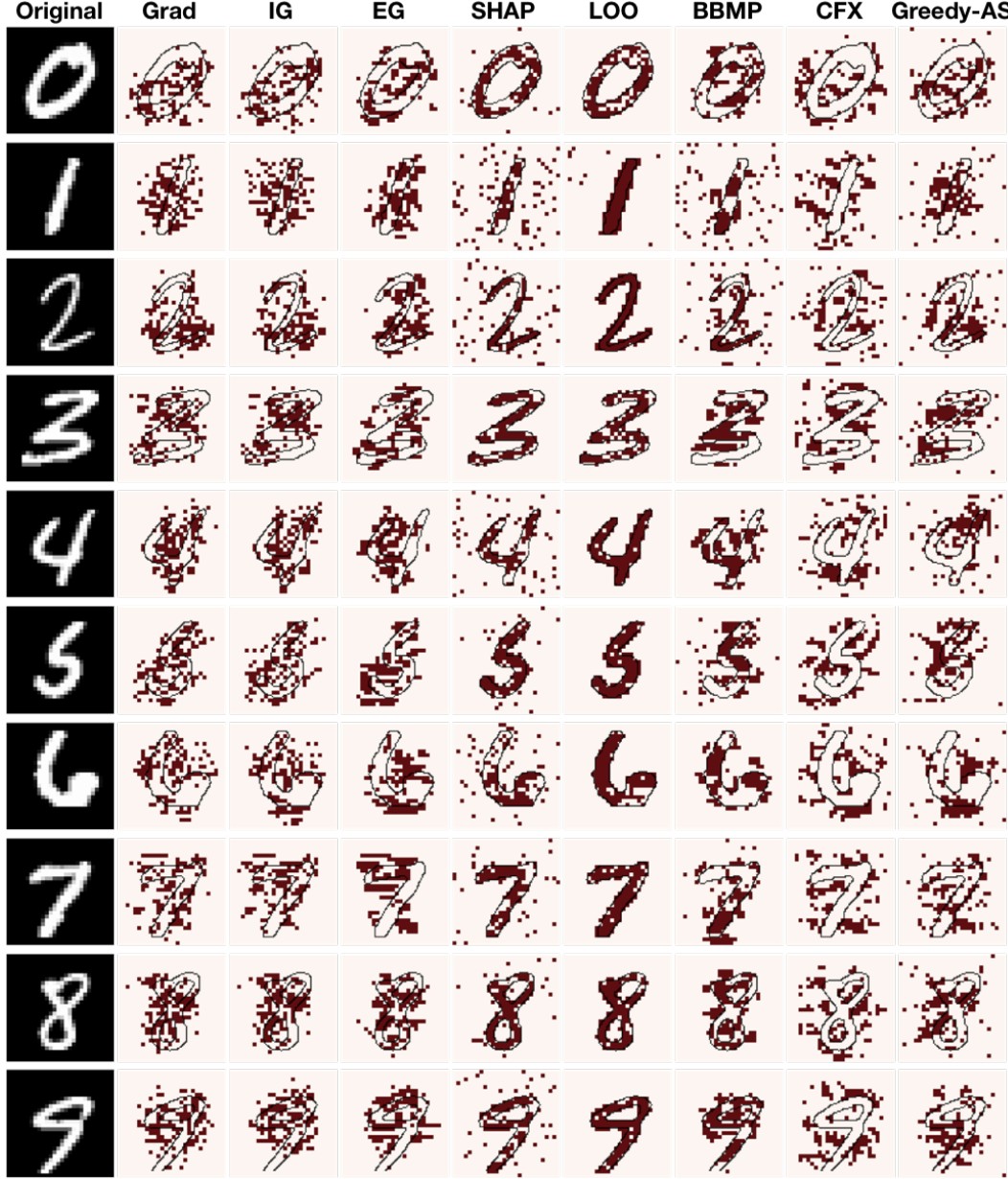

Figure 18: Visualization of different explanations on MNIST. The highlighted pixels are the top 20% relevant pixels selected by different methods. We see that Greedy-AS focuses both on crucial positive and pertinent negative regions.

## L    VISUALIZATION ON IMAGENET

We provide more examples on visualized explanations on ImageNet in Figure 19.

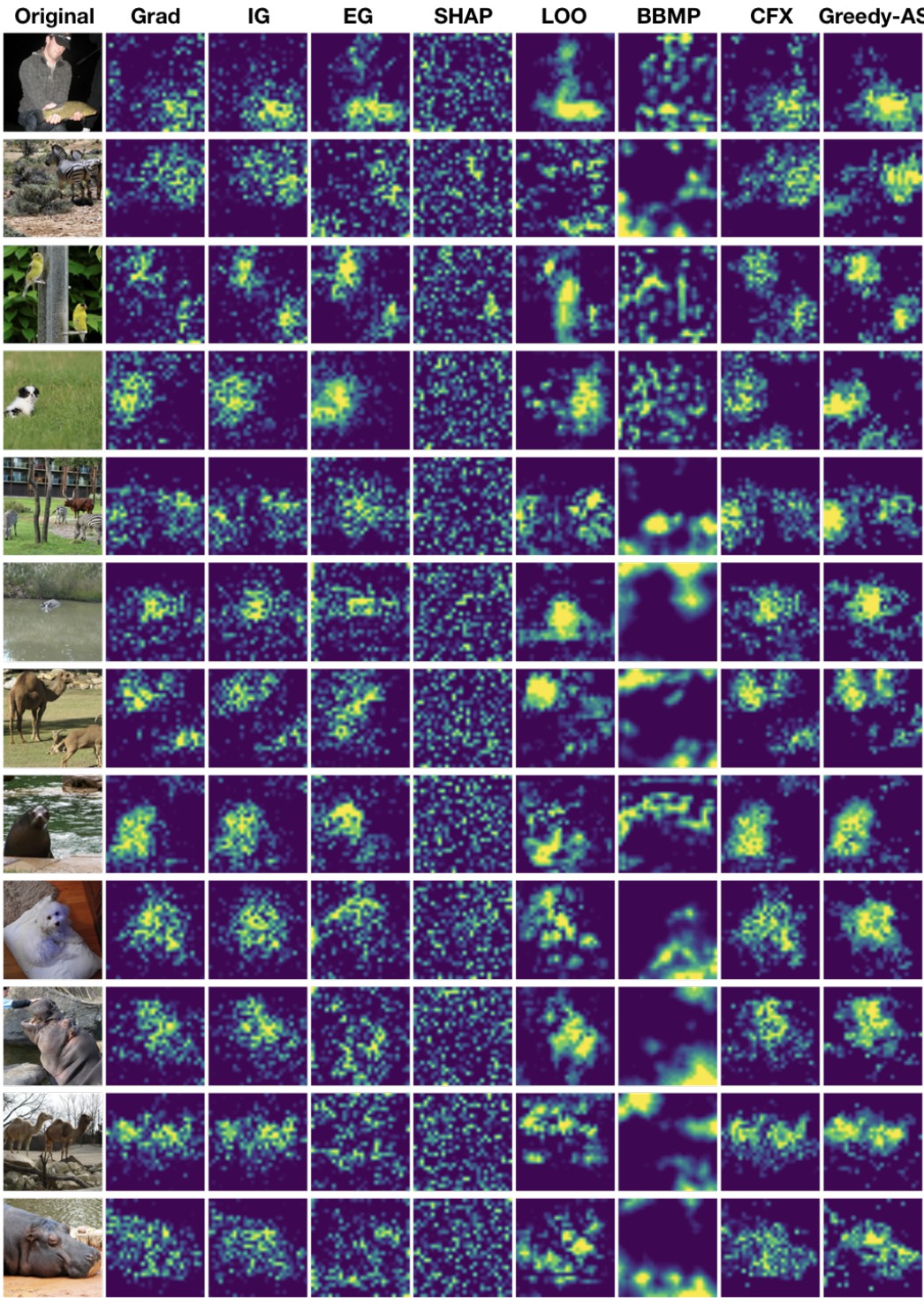

Figure 19: Visualization of different explanations on ImageNet. We see that Greedy-AS focuses more compactly on the areas containing the actual objects being classified.

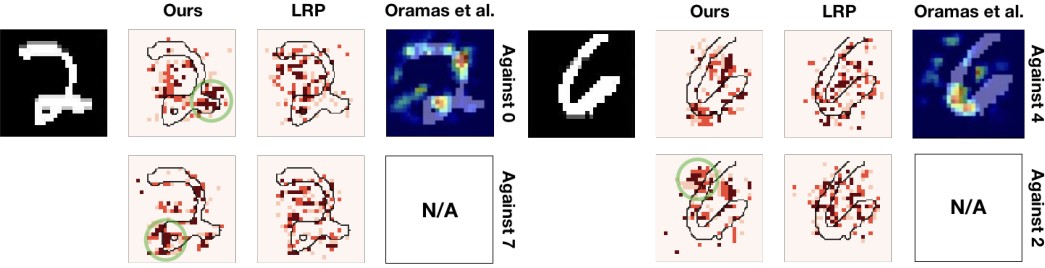

(a) Targeted explanation for 2 against 0 and 7 respec-  (b) Targeted explanation for 6 against 4 and 2 respec-
tively.  tively.

Figure 20: Comparisons between different methods for targeted explanation against different target classes on MNIST. We note that Oramas et al. (2019) could not handle arbitrarily specified target class and thus is not available to produce different explanations for different target classes.

## M  COMPARISONS WITH OTHER EXPLANATIONS CAPTURING PERTINENT NEGATIVE FEATURES

While the capability of capturing not only the crucial positive but also the pertinent negative features have also been observed in some recently proposed explanations. Most current methods are not explicitly designed to handle the targeted explanation task which attempt to answer the question "what are the important features that lead to the prediction of class A but not class B", and thus has different limitations. For example, the ability of LRP Bach et al. (2015) to capture pertinent negative features heavily depends on the input range. In Samek et al. (2016), the input images are normalized to have zero mean and a standard deviation of one where the black background will have non-zero value. In such case, LRP would have non-zero attributions on the black background pixels which allows the explanation to capture pertinent negative features. However, as later shown in Dhurandhar et al. (2018), if the input pixel intensity is normalized into the range between 0 and 1 where the background pixels have the values of 0, LRP failed to highlight pertinent negative pixels since the background would always have zero attribution. This is due to the fact that LRP is equivalent to multiplication between Grad and input in a Rectified Linear Unit (ReLU) network as shown in Ancona et al. (2018). In Oramas et al. (2019), unlike our targeted explanation where we know exactly which targeted class the explanation is suggesting against (and by varying the targeted class we observe varying corresponding explanation given), their method by design does not convey such information. The pertinent negative features highlighted by their method by construction is not directly related to a specific target class, and users would need to infer by themselves what target class the pertinent negative features are preventing against. To further grasp the difference, we compare our explanation with theirs in Figure 20. We borrow the results from Oramas et al. (2019) for visualization of their method. Qualitatively, we observe that our method gives the most intuitive explanations. For example, in Figure 20a, the first row shows different explanations that highlight different relevant regions supporting the prediction of a 2 instead of 0. Among the explanations, our method is the only one that highlights the right tail part (green circled) of the digit 2 which also serves as crucial evidence of 2 against 0 in addition to the left vertical gap (which when presented would make 2 looks like a 0) that is roughly highlighted by all three methods. Furthermore, as we change the target class to 7 (as shown in the second row), the explanation provided by LRP does not seem to differ much. This suggests that LRP is not sensitive to the target class change and might fail to provide useful insights towards targeted explanation. On the contrary, we observe that our explanation has a drastic change between different target classes. When explaining against the target 7, our explanation highlights the lower left green circled part which when turned off will make 2 becomes a 7. These results might suggest our method is more capable of handling such targeted explanation task.

# N QUANTITATIVE RESULTS ON TEXT DATASET

In this section, we conduct a set of quantitative analysis on different explanations for text classification model on Yahoo!Answers dataset.

Table 9: AUC of the different evaluation criteria for various explanations on Yahoo!Answers. The higher the better for Robustness-$\overline{S_r}$ and Insertion; the lower the better for Robustness-$S_r$ and Deletion.

| Criteria | Grad | IG | SHAP | LOO | EG | Random | Greedy-AS | CFX |
|---|---|---|---|---|---|---|---|---|
| Robustness-$\overline{S_r}$ | 1.97 | 1.86 | 1.81 | 1.74 | 1.96 | 1.71 | **2.13** | 1.95 |
| Robustness-$S_r$ | 2.91 | 3.14 | 3.34 | 4.04 | 2.99 | 7.64 | **2.41** | 2.96 |
| Insertion (random baseline) | 0.06 | 0.06 | 0.07 | 0.18 | 0.20 | 0.10 | **0.21** | 0.05 |
| Deletion (random baseline) | 2.57 | 2.96 | 2.23 | 2.07 | 2.07 | 2.63 | **1.56** | 2.35 |
| Insertion (zero baseline) | 2.59 | 6.31 | **8.63** | 3.61 | 3.34 | 0.34 | 3.64 | 2.95 |
| Deletion (zero baseline) | 5.28 | 5.64 | **1.52** | 3.81 | 5.12 | 6.30 | 4.60 | 5.83 |

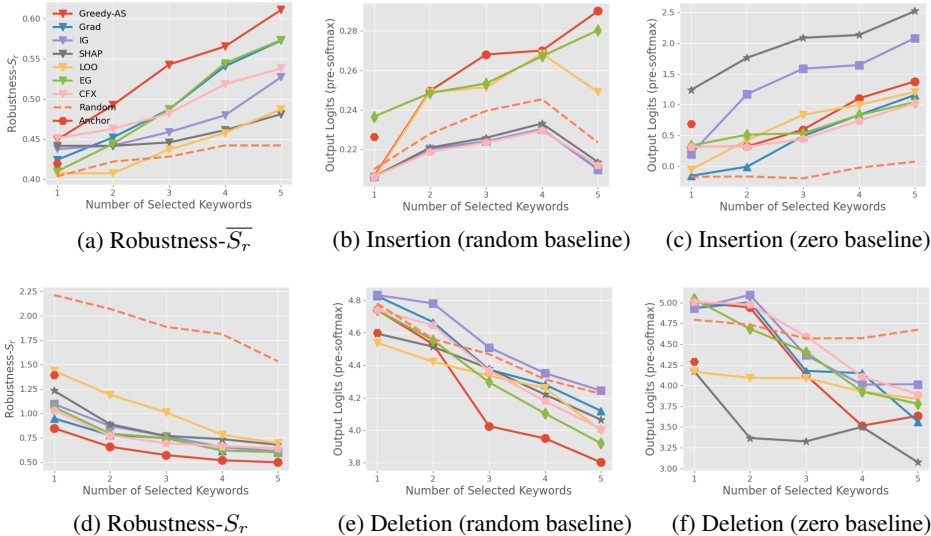

(a) Robustness-$\overline{S_r}$    (b) Insertion (random baseline)    (c) Insertion (zero baseline)

(d) Robustness-$S_r$    (e) Deletion (random baseline)    (f) Deletion (zero baseline)

Figure 21: Evaluation curves on Yahoo!Answers dataset.

# O    QUALITATIVE RESULTS ON TEXT DATASET

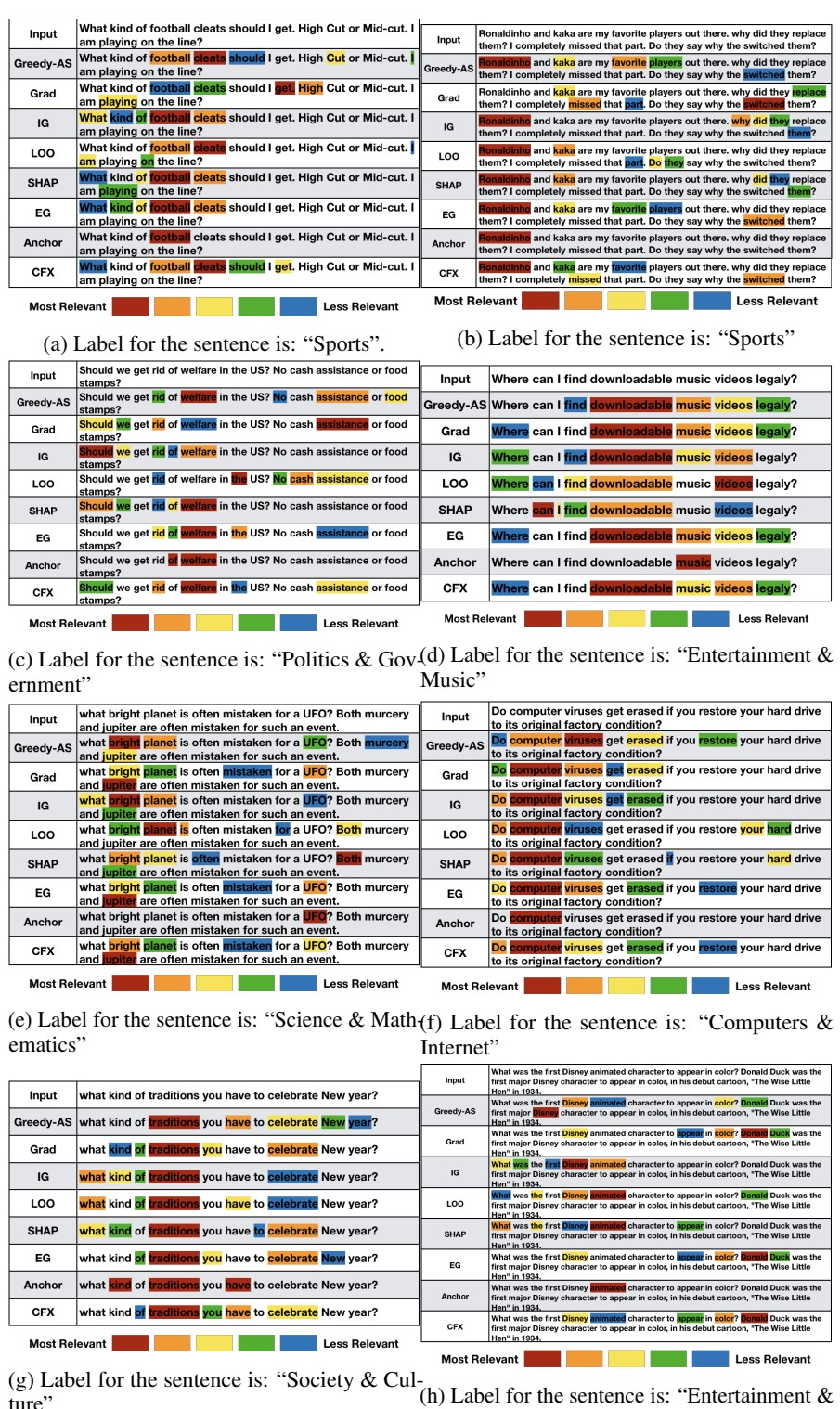

Figure 22: More examples for explanations on text classification model.

## P    USER STUDY ON TEXT DATASET

To further compare the top-5 relevant keywords provided by various explanations, we conducted an user study involving 30 different text examples (sentences) and 30 different users with Computer Science background by convenience sampling. In the study, each user is presented with a set of sentences and their corresponding categories. We then asked each user to highlight 5 to 7 keywords for each sentence that they consider to be most relevant to the ground truth label of the corresponding sentence. After collecting the feedback, for each sentence, we have 10 sets of different labels provided by the users. For each sentence, we then select the keywords that have the highest frequency of being marked among the users and treat those keywords as the ground truth explanation for the corresponding sentence.

We leverage such user labels to calculate the precision@k and mean Average Precision (mAP) for each explanation method. We show the results in Table 10 [5]. We observe that Greedy-AS has the highest overall precision score, while the top-1 keyword selected by Anchor and IG match user intuition the most. If we believe that users can correctly pick the correct keywords that are responsible for the model prediction, then the p@k and mAP measure how well the explanation captures the correct keywords that are responsible for the model prediction. We believe the good result on user study provides extra evidence that our method identifies features that are responsible for the actual prediction.

Table 10: Precision of Explanations with User Labeled Ground Truth

|  | p@1 | p@2 | p@3 | p@4 | p@5 | mAP |
|---|---|---|---|---|---|---|
| Greedy-AS | 0.83 | **0.78** | **0.78** | **0.75** | **0.68** | **0.76** |
| Grad | 0.83 | 0.70 | 0.72 | 0.67 | 0.61 | 0.71 |
| IG | 0.83 | 0.67 | 0.56 | 0.52 | 0.50 | 0.61 |
| SHAP | 0.73 | 0.75 | 0.61 | 0.58 | 0.51 | 0.64 |
| LOO | 0.67 | 0.67 | 0.52 | 0.48 | 0.45 | 0.56 |
| EG | 0.90 | 0.75 | 0.69 | 0.67 | 0.63 | 0.73 |
| Anchor | **0.93** | - | - | - | - | - |
| CFX | 0.83 | 0.68 | 0.67 | 0.61 | 0.57 | 0.67 |
| Random | 0.17 | 0.27 | 0.28 | 0.29 | 0.27 | 0.25 |

## Q    RUNTIME ANALYSIS

We show below the average runtime (wall clock time) of different methods for computing explanation for a single image on MNIST and ImageNet. For Greedy and Greedy-AS, we show the time needed to compute the top-20% relevant features.

Table 11: Runtime of Different Explanations.

| Explanation | Grad | IG | LOO | BBMP | SHAP | CFX | EG | Greedy | Greedy-AS |
|---|---|---|---|---|---|---|---|---|---|
| MNIST Runtime (second) | 0.005 | 0.023 | 0.004 | 4.154 | 9.673 | 8.582 | 6.123 | 9.532 | 13.621 |
| ImageNet Runtime (second) | 0.024 | 0.138 | 0.849 | 8.536 | 111.794 | 25.577 | 64.270 | 117.781 | 135.838 |

---

[5]Note that as Anchor tends to select only one keyword in each sentence, we provide only its precision@1 in Table 10.

