# OpenReview forum: "Evaluations and Methods for Explanation through Robustness Analysis"
_ICLR.cc/2021/Conference — ICLR 2021 Poster_

### Official Review · AnonReviewer3 · 2020-10-20
**An effective explanation method based on robustness**

**Rating:** 7
**Confidence:** 4

**Review:**

The paper proposes an effective explanation method based on a notion of robustness defined by the authors.
The paper is well presented and easy to follow. It compares against state-of-the-art methods and provides valid and statistically significant experimentation. The explanations returned are different from the competitors and useful.

However, some points that should be addressed before publication.
First, I suggest to add the evaluation measures for robustness presented in
Hara, S., Ikeno, K., Soma, T., & Maehara, T. (2018). Maximally invariant data perturbation as explanation. arXiv preprint arXiv:1806.07004.
The evaluation of Hara is similar to the one adopted w.r.t Insertion/Deletion.
In addition, an evaluation measure for robustness different from everything presented in the paper but quite important for having a different validation is the one proposed in
Melis, D. A., & Jaakkola, T. (2018). Towards robust interpretability with self-explaining neural networks. In Advances in Neural Information Processing Systems (pp. 7775-7784).
based on the local Lipschitz estimation.

Second, an explanation similar to the one reported in the paper is returned by the method presented in
Guidotti, R., Monreale, A., Matwin, S., & Pedreschi, D. (2019, September). Black Box Explanation by Learning Image Exemplars in the Latent Feature Space. In Joint European Conference on Machine Learning and Knowledge Discovery in Databases (pp. 189-205). Springer, Cham.
Another method that should be considered for comparison as it is based on robust anchors is the one presented on:
Ribeiro, M. T., Singh, S., & Guestrin, C. (2018, February). Anchors: High-Precision Model-Agnostic Explanations. In AAAI (Vol. 18, pp. 1527-1535).
It would be interesting to compare the Greedy proposals with these two methods both quantitatively and qualitatively.

Third, an important aspect that is missing for the model proposed is either an analysis of the complexity or even better, the running time for returning explanations. In this way, the user can understand which is the best compromise between robustness and speed.

Minor issues:
- The results on textual data are not as convincing as those on images. The authors should treat this part of the presentation carefully and with more attention.
- The authors do not specify in the main paper (or it is not easy to find) which are the classification models explained.

---

> ### Author Response · Authors · 2020-11-21
> **Response to Reviewer 3**
>
> Thank you for pointing us to the many interesting studies. We will include discussions with each of the mentioned work in our revision.
>
> Q1: I suggest to add the evaluation measures for robustness presented in Hara, S., Ikeno, K., Soma, T., & Maehara, T. (2018).
>
> A1: The evaluation utilized in (Hara et al., 2018) is indeed very similar to the Insertion criterion presented in our study. In fact, we could view the Insertion criterion as an opposite-order version of the criterion adopted by Hara et al. Specifically, Hara et al. begin by masking the most irrelevant t% of features and measure the prediction change. This actually corresponds to the Insertion criterion when we have inserted (100-t)% of the relevant features (and t% of the irrelevant features are masked). Thus, the evaluation in Hara et al. should be strongly correlated with the presented Insertion criterion. There is however one implementation difference where Hara et al. model feature absence by setting the masked region of the image to gray pixels while we set the values randomly. In revision Appendix L, we include additional experimental results on setting the masked pixels to 0.5 which corresponds to the gray pixels adopted by Hara et al.
>
> Q2: Evaluation based on local Lipschitz estimation proposed by  Melis, D. A., & Jaakkola, T. (2018).
>
> A2: We agree it would be interesting to evaluate our method on the notion of local Lipschitz estimation, or more generally the "sensitivity" of the explanation [1], which estimates the lower bound of the local Lipschitz by sampling (without the black-box optimization). We have included a sensitivity analysis result in revision Appendix Q.
>
> [1] On the (in)fidelity and sensitivity for explanations. Chih-Kuan Yeh, Cheng-Yu Hsieh, Arun Sai Suggala, David I. Inouye, and Pradeep Ravikumar. NeurIPS 2019.
>
> Q3: Additional baselines: (Ribeiro et al., 2018) and (Guidotti et al., 2019)
>
> A3: We have initially included Anchor (Ribeiro et al., 2018) into our baseline on the text classification results shown in Figure 17 and 18. We note that unlike other feature attribution methods that generates an importance score to each feature, Anchor aims at selecting a shortest set of keywords that when anchored could yield the highest precision. As a result, for many examples shown, it only selects relatively few keywords (but also the important ones) compared to other methods, and thus we cannot calculate the whole Insertion/Deletion/Robustness-$S_r$ ($\\overline{S_r}$) score (but we plot one point, corresponding to its top-1 keyword performance, in Figure 17). We have also included discussion with (Guidotti et al., 2019) in our revision. Similar to the proposed method, the explanation presented in (Guidotti et al., 2019) is able to highlight both areas that contribute to the current prediction and areas that contrast the prediction by learning image exemplars and counter-exemplars in the latent feature space. The method however relies on an effective generative model which might not be easily obtainable in different applications.
>
>
> Q4: An analysis of the complexity or even better, the running time for returning explanations. In this way, the user can understand which is the best compromise between robustness and speed.
>
> A4: We provide in Appendix P an initial study on the runtime analysis for different methods. We show the average runtime (wall clock time) of different methods for computing explanation for a single MNIST image. For Greedy and Greedy-AS, we show the time needed to compute the top-20% relevant features.
>
> We see that LOO has the least running time since its computation can be done with a single batch forward pass to the model. Gradient-based methods are also fast since they only need constant times of forward and backward passes of the model. Optimization based methods like BBMP and CFX relatively take longer time as they require multiple steps of gradient descent. SHAP as well requires longer runtime as it depends on sampling and solving a kernel regression. Comparing to existing explanations, our method indeed is not as fast since we require running PGD for multiple candidate sets to evaluate their robustness in each greedy step. We could however speed out the process by leveraging GPU to parallely perform PGD on multiple candidates simultaneously and further reduce the computational time by relaxing PGD constraints such as performing less descent steps (with the trade-off of less precise robustness value) and selecting multiple features at each greedy step. We note that the time reported is measured with a single GPU, and further speed-up is available by parallelizing the computation over multiple GPUs. We will include detailed runtime comparison over different methods on different datasets in the final revision.

---

> > ### Author Response · Authors · 2020-11-21
> > **Response to Reviewer 3 (part 2)**
> >
> > Q5: The results on textual data are not as convincing as those on images.
> >
> > A5: We have expanded the experiments on textual data in our initial revision by including comparisons with more baselines both qualitatively and quantitatively in Appendix M and N. Quantitatively, we observe similar trends as the ones in image datasets. Qualitatively, it appears to us that our method tends to select keywords that are more closely related to the ground truth category of the presented sentence. We are also working on expanding our user study in Appendix O by involving more participants and examples to validate the result.
> >
> >
> > Q6: The authors do not specify in the main paper (or it is not easy to find) which are the classification models explained.
> >
> > A6: We have clarified the experiment setup in the initial revision Section 5, and the detailed setting is provided in Appendix C.

---

### Official Review · AnonReviewer4 · 2020-10-28
**Interesting paper focusing on evaluating explanations through robustness analysis, using adversarial perturbations. Doubts about the basis for that and the "human" study. I dont know if another ML/AI will be more suitable for this paper than ICLR..**

**Rating:** 6
**Confidence:** 3

**Review:**

The main contribution of this work is the proposal, implementation and validation of a new evaluation criteria/measure for explanations based on robustness analysis. The main idea, as I understand it, is innovative, adversarial perturbations: other feature explanation solutions rely on removing features, the method here presented learns a set of features so that we get the smallest perturbation that change in outcome for the selected set and largest perturbation that changes outcome for the rest of the features.

I think that interpretable ML and  Explainable AI need more work on evaluation, and providing measures for that is relevant in these areas. So, I wonder if other conferences in ML/AI will benefit more of this work than ICLR. In general I find the paper strong, interesting and innovative, technically sound.

The paper provides a very strong empirical analysis using various datasets, even if for me it was hard to follow all the figures, explanations, methods Greedy/Grad. It is hard to interpret some of the figures, fig 2 and 3 provided, since we need to analyze pixels, and check of they are “less noisy”, “rather noisy”, “noisy” etc….

I am not very familiar with using “human study” for a user evaluation, use user study. “Human” study sounds odd to me.

I was hoping that the user study announced in the abstract and introduction would validate the presented metric, and the methods studied. This is really important, since we make explanations to people. I was disappointed to see that the so called “human study” is described in just few lines, telling that 10 people (without specifiying anything else, what kind of participants were those?) labelled the sentences to show that Greedy matches best their labelling (I guess that 5 people per Grad/Greedy). It is hard to see how this user study strengthens the paper, what is the difference between this evaluation and the other presented in the paper were other “humans” observe things on the data/images? It looks to me that the user study is an afterthought, a good attempt to complement the paper, but either is properly done or removed.


But one of my main concerns is actually is the proposed solution, like the actual motivation or basis for it, can they really identify features that are responsible for the predictions made or just find regions that have high sensitivity to changes (adversarial perturbations)? It might be that I got lost in all the evaluations, and I cannot figure out if the real features the model used to make the prediction will be found, or just those that make the prediction change…. Is that so? Are those the same features?


While reading, I found some small grammar issues; please, double-check the text.

---

> ### Author Response · Authors · 2020-11-21
> **Response to Reviewer 4**
>
> Q1: I am not very familiar with using “human study” for a user evaluation, use user study. “Human” study sounds odd to me.
>
> A1: We agree that "user study" is more appropriate for the context. We have revised the terminology in our revision.
>
> Q2: I was hoping that the user study announced in the abstract and introduction would validate the presented metric, and the methods studied. It is hard to see how this user study strengthens the paper, what is the difference between this evaluation and the other presented in the paper where other “humans” observe things on the data/images?
>
> A2: While we are currently working on expanding the user study with more users and baselines involved, we first clarify the details of the original study reported in the paper. In the original study, we show 10 different users with computer science background the same set of text examples with their corresponding categories (examples shown below), and we then ask the users to highlight 5 to 7 keywords they consider to be most relevant to the example’s corresponding category. We note that this step of labeling keywords is model-agnostic, so we do not split users for different methods being evaluated and thus their choice does not suffer from confirmation bias. For each example, After collecting the feedback, we have 10 different sets of highlighted keywords for each example respectively. For each example, we then select 5 keywords that have the highest frequency of being highlighted by the users, and treat the keywords as the corresponding explanation ground truths for the example. With the ground truths, we calculate the p@k and mAP for different explanations. We note that the user studies measure how much the top keywords selected by each explanation matches the top keywords selected by users. If we believe that users can correctly pick the correct keywords that are responsible for the model prediction, then the p@k and mAP measures how well the explanation captures the correct keywords that are responsible for the model prediction. We believe that the good result on user studies provides extra evidence that our method identifies features that are responsible for the actual prediction, not just the sensitive features.
>
> In the initial rebuttal, we included much more methods in our user study in Appendix O (note that human annotation is method-agnostic, so we do not need to collect additional human information to add new methods). We will further include a more comprehensive and detailed discussion on the undergoing user study (with more users and more examples) into our revision by the end of the rebuttal period. We believe that the user study is a key component of our paper, and we will largely improve the quality of it before the end of the rebuttal period.
>
> Example texts and their corresponding categories shown to the users:
> - How do you put a password on a computer folder, or is there a free program that does it? (Computers & Internet)
> - I contribute 13% of my paycheck to my 401K. Is that enough if my salary continues to increase each year? I work in the IT industry. (Business & Finance)
>
> Q3: One of my main concerns is actually is the proposed solution, like the actual motivation or basis for it, can they really identify features that are responsible for the predictions made or just find regions that have high sensitivity to changes (adversarial perturbations)?
>
> A3: We emphasize that our method also captures features that when anchored, makes the model robust. These features are not necessarily highly sensitive to change. For example, $f(x_1, x_2) = 0.8 * sign(x_1-1) + 0.3 sin(x_2)$, and we want to explain why $f(x_1,x_2) > 0.5 at (2,2)$. We note that when $x_1 > 1$, $f(x_1,x_2) >= 0.8 - 0.3 = 0.5$. However, around $(2,2)$, clearly $x_2$ is much more sensitive than $x_1$ (one evidence is that the gradient of $x_1$ is 0 and the gradient of $x_2$ is $0.3 cos(x_2)$ ), but the feature which makes the prediction robust after anchored is $x_1$. We hope this example shows that our method finds features that are responsible for prediction (such as $x_1$) instead of features that have high sensitivity to changes (such as $x_2$).
>
> Q4: I found some small grammar issues; please, double-check the text.
>
> A4: We have conducted an initial round of overall polishing of the paper, and will further revise it for the final draft.

---

> > ### Comment · AnonReviewer4 · 2020-11-23
> > **brief note**
> >
> > Thanks for your clarifications and for making adjustments. I think I do understand now better my concern about the actual basis for the solution presented. I think the paper is above the acceptance level.
> >
> > Best, reviewer

---

### Official Review · AnonReviewer1 · 2020-10-29

**Rating:** 7
**Confidence:** 3

**Review:**

Many interpretability techniques center on identifying the subset of “most relevant features”. In this work, the authors propose defining this set as the set of features that are easiest to attack adversarially (in an $L_p$ sense).

First off, the paper was somewhat difficult to read because of a frustrating amount of vertical space LaTeX hacking, so much so that the spacing between sections and paragraphs is even less than the normal spacing between sentences. This is not a good way of squeezing everything into 8 pages.

Apart from that, the paper is comprehensive in its experimental evaluation, with a good suite of appropriate baselines, sanity checks, and human studies. I think that it is an interesting complement to the current suite of feature attribution techniques. Conceptually, it is quite similar; as the authors note, there have been many related techniques that try to “remove features” by adding noise to them, setting them to a baseline value, or blurring them; here, the authors instead consider adversarially perturbing them, and they propose a modified greedy strategy that seems to work well. It is a bit unclear to me why considering adversarial perturbations is a priori any more convincing than, e.g., considering blurring out or adding noise to the selected features, but they serve slightly different purposes and it's reasonable that these methods could be complementary.

Empirically, this method leads to gains under insertion and deletion metrics, compared to existing techniques. I had a couple of questions about these baseline comparisons:

1. I’m a bit confused as to why BBMP does so poorly under the insertion and deletion games, given that it explicitly optimizes for these. Could the authors discuss this? From Figure 11, it looks like the implementation of BBMP seems unoptimized; at least, it returns degenerate results that are very different from the examples shown in the original paper. There's a follow-up https://arxiv.org/pdf/1910.08485.pdf that similarly optimizes for a mask of a fixed size, which seems to outperform many of the other methods here, so I'm confused why the results are so different.

2. S5.2: The authors say that SHAP obtains a better performance under the deletion criterion because it exploits adversarial artifacts. Shouldn’t this problem be even worse for the Greedy(-AS) methods, which explicitly seek to exploit adversarial artifacts? Why are we not seeing that?

**Details:**

3. Assumptions 1 and 2 are not precisely defined; please either state them formally, or make it clear that they are informal motivations.

4. Definition 3.1: is $\epsilon_S^*$ a function of $x$ and $y$? How are the overall Robustness values calculated then? By averaging?

**Update**
Thank you to the authors for the detailed responses and revisions. I have adjusted my score upwards. I think this is a useful exploration of an alternative means of quantifying feature importance, with intriguing results: somehow, optimizing for adversarial robustness also seems to optimize for the score on insertion/deletion games. Further exploration of that issue (or at least, elevating some of the many appendices on that issue to the main text) would, in my opinion, increase the impact of the paper.

---

> ### Author Response · Authors · 2020-11-21
> **Response to Reviewer 1**
>
> Q1:  It is a bit unclear to me why considering adversarial perturbations is a priori any more convincing than, e.g., considering blurring out or adding noise to the selected features, but they serve slightly different purposes and it's reasonable that these methods could be complementary.
>
> A1:  We discuss one caveat for using reference value explanations and evaluations respectively -- if the feature value equals the reference value, the feature will not be deemed as important (even if it is the key to prediction) by the explanation method as well as the evaluation. We added simple theoretical analysis to justify the intuition in Appendix A.
>
> In Appendix A.1, we added analysis which show that many explanations are biased to reference value (IG, SHAP, LRP, DeepLift). For example, if the reference value $x'$ is a zero vector (which means black pixels in image), then any black pixel will get a zero attribution value no matter if the object of interest is actually back. We note that the Expected Gradient (Erion et al 2019) is not biased to reference value by this theoretic definition since the baseline $x’$ is actually a distribution. However, for feature values $x_i$ that are close to the distribution of $x'_i$, the attribution score will be lower (but not 0 as our theoretic definition), when feature values $x_i$ are far from the distribution of $x'_i$, the the attribution score will be larger, which still has some biased involve. We leave further investigation of the problem to future work to use more advanced analysis to quantify such a bias for explanations when the baseline follows a distribution.
>
> In Appendix A.2, we added theoretical analysis that when a feature is equal to the replaced reference value, no matter how important it actually is, it will not contribute to the Deletion score nor the Insertion score. However, the lower the deletion score is better, and the higher the insertion score is better, and choosing an important feature that corresponds to the reference value will inevitably not improve the Deletion score and Insertion score. The reference values such as blurring out or adding noise may still make the original features unchanged (such as when the main part of the image is already blurred, blurring the image will not change the image, and thus the main part of the image is biased to blurred baseline).
>
> Q2: I’m a bit confused as to why BBMP does so poorly under the insertion and deletion games, given that it explicitly optimizes for these. Could the authors discuss this?
>
> A2: One possible explanation for the suboptimal performance of BBMP under the insertion and deletion games is the mismatch between the blurring baseline implicitly adopted by BBMP and the random value baseline used in our implementation of Insertion and Deletion criteria. In addition, BBMP includes a TV term that regularizes the generated mask to be smooth aside from focusing only on decreasing (or increasing) the prediction output. This could also potentially restrict its aim on optimizing for the Insertion and Deletion game. We have worked on further finetuning the hyperparamters for BBMP on ImageNet and observed improved results of BBMP on the Insertion criteria (see revision Table 2 and Figure 9b). Specifically, we lower the TV regularization term to obtain better results. We have included the update in the revision.

---

> > ### Author Response · Authors · 2020-11-21
> > **Response to Reviewer 1 (part 2)**
> >
> > Q3: The authors say that SHAP obtains a better performance under the deletion criterion because it exploits adversarial artifacts. Shouldn’t this problem be even worse for the Greedy(-AS) methods, which explicitly seek to exploit adversarial artifacts? Why are we not seeing that?
> >
> > A3: In revision Figure 16, we investigate further into the reason why seemingly noisy explanations provided by SHAP could achieve strong performance under the Deletion criteria on ImageNet. To look into the problem, we introduce a random explanation baseline where the importance score for each feature is uniformly sampled from [0, 1]. Such random explanation is totally independent of the model and input, and therefore shouldn’t be regarded as a good explanation. However, when we evaluate such random explanation under the Deletion criterion, we observe that the random explanation is much closer to compared methods than the insertion score, even outperforming half of the methods when the reference value is set to 1. The result somewhat suggests that the Deletion criterion might not be suitable for evaluating explanations on high-dimensional inputs, as even random noise could exploit the artifacts to achieve a high score. Such a phenomenon is less severe with the proposed $\\text{Robustness-}S_r$ criterion (which resembles the Deletion criterion), as all methods that generate explanations focusing on the true object for classification (e.g., IG/Grad) outperforms the random explanation baseline by a large margin, as shown in Figure 8b.  We further note that even though our Greedy-AS algorithm exploits regions that are most suspect to adversarial attacks, the regions that are suspect to attacks themselves may still be meaningful as shown in our visualization result.
> >
> > Q4: Assumptions 1 and 2 are not precisely defined; please either state them formally, or make it clear that they are informal motivations.
> >
> > A4: We have clarified the assumptions as more of an informal motivation in the revision.
> >
> > Q5: Definition 3.1 $\epsilon^*_{S}$ is a function of $x$ and $y$? How are the overall Robustness values calculated then? By averaging?
> >
> > A5: Definition 3.1 corresponds to the robustness value on a single input and $g$ is indeed a function of  the model $f$, the input $x$ and the predicted class $y = f(x)$. To calculate the overall robustness over multiple data points, we take the average value following literature in adversarial robustness. More formally, for a set of testing data points of interest $\\{x_i, y_i=f(x_i)\\}_{i=1}^n$ and their corresponding relevant sets generated by an explanation $\\{S_i\\}_{i=1}^n$. We calculate the average robustness as $\\frac{1}{n} \\sum g(f, x_i, S_i)$. We have made clarifications in the revision.

---

### Official Review · AnonReviewer2 · 2020-10-30
**Good ideas for explanations and benchmarking, needs more motivation and more detailed evaluation**

**Rating:** 7
**Confidence:** 4

**Review:**

Summary: The goal of the paper is to present a new explanation benchmark and explanation method for ML models. I think the benchmark is interesting but insufficiently motivated, especially given existing concern in the field about the degree to which small local perturbations yield biased results because of saturation. The explanation method seems to work well, even on removal-based benchmarks, which is great! Evaluation is limited w.r.t. datasets and methods compared against, however. Overall, some serious positives and important negatives make my impression borderline, leaning reject. However, addressing the weaknesses well could definitely raise my score.

**Update**: The authors' update is comprehensive, well-thought out, and demonstrates significant improvements to the paper; I have raised my score to reflect this.

Objective: Use the adversarial robustness framework to develop both a benchmark for ML model explanations and new methods to actually explain ML models.

Strengths:
* I think the distinction from previous removal-based work, which to my knowledge is the primary benchmarking focus, is well-posed.
* The adversarial robustness framework is presented very clearly and seems to get at a different question than previous benchmarks, which is a useful contribution.
* The explanation method seems well-designed.
* Experiments show very good results for Greedy-AS.
* I really appreciated Table 2, because it does not seem like enough to show Greedy-AS does the best job of optimizing for the thing (only) it optimizes for. These additional results are impressive, though I wish I understood why Greedy-AS does better on them.

Weaknesses:
* From the introduction, I am a little concerned about this method because of its intentional focus on small perturbations. This is a well-known problem with gradients that integrated gradients improves on by adding a reference; gradients measure small perturbations around the input, but do a bad job detecting important features because nonlinearities often saturate around the inputs. I think there's a lot of empirical evidence that IG improves on grads for this reason. The benchmark is subject to this issue -- I tend to think a priori that a test where grads shows good performance may be a flawed test -- and I'd tend to think the explanation method would suffer for this reason as well. Overall, it is known that explanation methods exist that do not need references; these just tend to be worse on a lot of benchmarks, for reasons like saturation. There's a lot of room here to boost the value of the paper, in my opinion. (1) Explain why saturation isn't an issue with this method and (2) Tie the good results with Greedy-AS to what seems to be the key motivation for its introduction -- the independence of reference. If you could show that IG, etc are giving worse results *because* they're being biased by their reference, it'd be very helpful.
* It makes sense to me that Greedy-AS does better on the benchmark that inspired it, but why does it do better on the removal-based methods? In some sense, this is exactly what methods like SHAP should be optimizing for, so the biggest drawback to this result (which is a good result for your method!) is that I don't have an intuition for why it should be the case. I'd also be interested to see a comparison with a method like Expected Gradients (Erion et al 2019, Sturmfels et al 2020), which averages IG-type explanations over multiple references by default. I also would like to see results on a low-dimensional, maybe tabular dataset, because when SHAP is entirely sampling-based it will have trouble with large numbers of pixels. I'd also be interested in seeing the SHAP results on text. I think it may do a lot better in that case. Even if SHAP wins on tabular or text data, this could be a good thing for your paper, because then it'd be clearer under what conditions Greedy-AS works well.
* A related concern is the choice of explanation methods in the experiments; for example, IG is one of the best performers in the Table 1/2 benchmarks, but is not used in Figures 4 and 5. Some justifications for the methods picked in these figures would be good; right now it seems like some of the lower-performing methods are being displayed while ideally the strongest benchmarks would be displayed. I think some justification is required here before publication.
* I'm actually more sold on the explanation method than on the benchmark; I think there's not quite enough justification for the benchmark aside from "other methods are biased by the reference." However, there's not much empirical or theoretical basis for believing the use of a reference is intrinsically bad (as discussed above, references may induce bias, but small perturbations can be biased by things like saturation). I know it's hard to show why one benchmark is better than another, but if the benchmark is a key contribution I think I need more analysis on why it's a better benchmark.
* I'd also appreciate a bit more detailed results on the non-MNIST experiments, to convince me the method works well in many settings.
* Overall, having more non-image experiments, and associated discussion, could be helpful. Right now almost all results are images, with one small text example, and I wouldn't be surprised to see very different results on different types of data.
* I'm concerned the method could be very slow, given that the adversarial optimization is itself embedded in a sampling-based procedure for the Greedy-AS method. Can you add some discussion of the runtime?

---

> ### Author Response · Authors · 2020-11-20
> **Response to Reviewer 2 (part 1)**
>
> Q1:  Concerns about the proposed method because of its intentional focus on small perturbations. Explain why saturation isn't an issue with this method.
>
> A1: We agree Grad which intentionally focuses on “infinitesimal” perturbations is subject to the saturation problem and provides rather local explanations. However, we note that saturation only happens when $f(x)$ and $f(x’)$ ($x$ after the perturbation) are close. In Grad, $x’$ is defined as $x + \delta$, where $\delta$ is an infinitesimal perturbation around $x$; in IG, $x’$ refers to the baseline; and in our proposed method, $x’$ refers to the minimum perturbation so that $f(x’) \neq f(x)$ (for a more formal definition see Eq (1) in our paper). For IG,  if $f(x) \neq f(x’)$, they can resolve the saturation issue (see the premise of the  axiom sensitivity (a) in the IG paper). On the other hand, the adversarial perturbation we considered in this study will also not suffer from the saturation issue by design, since after the adversarial perturbation, even though $x$ and $x’$ might be close, $f(x)$ and $f(x’)$ will be different by the definition and thus the saturation issue will not occur.
>
> We provide an illustrating example below to show the distinction between our method and Grad, where Grad suffers from the saturation problem whereas our method could still successfully distinguish the importance among the input features. Consider a binary classification model that takes two-variable input $f(x_1, x_2) = sign(1 - ReLU(1 - x_1))$. Naturally, one would regard $x_1$ as relevant and $x_2$ as irrelevant since the classification result in fact only depends on $x_1$. However, for a given input $(x_1, x_2) = (2, 5)$, the explanation (attribution score) provided by Grad will be $(0, 0)$  (as the function becomes flat at $x_1 = 1$), which fails to distinguish between relevant and irrelevant features. However, in this case, our method could still distinguish between the relevance level of $x_1$ and $x_2$, as the $Robustness-S_r = 2$ if $S_r = \{x_1\}$ and infinity if $S_r = \{x_2\}$, suggesting $x_1$ is more important than $x_2$. Note that if the reference value for IG is $(x_1,x_2) = (1,1)$, then the explanation (attribution score) provided by IG will be $(0, 0)$, but if the reference value for IG is $(x_1,x_2) = (0, 0)$, then the explanation (attribution score) provided by IG will be (1, 0). Therefore, when the reference value in IG is not chosen correctly, they may also suffer from saturation.

---

> ### Author Response · Authors · 2020-11-20
> **Response to Reviewer 2 (part 2)**
>
> Q2: Tie the good results with Greedy-AS to what seems to be the key motivation for its introduction -- the independence of reference. If you could show that IG, etc are giving worse results because they're being biased by their reference, it'd be very helpful.
>
> A2: In the revision Appendix L, we conducted a detailed analysis on the effect of reference value. We anticipate that the inferior performance of IG and SHAP are due to their reference value being mismatched to the reference value of the evaluation (recall that for insertion score and deletion score, they replace the removed feature by random value, where the random value refers to the reference value of the evaluation). Our hypothesis is that when the reference value of explanation matches the reference value of the evaluation (insertion score and deletion score), then IG and SHAP will perform better. When they do not match, the IG and SHAP will perform worse due to the reference value mismatch (which can also be considered as biased by their reference since in practice we do not know which reference value makes the most sense for evaluation). To validate our hypotheses, we added multiple reference values (0, 0.25, 0.5, 0.75, 1) in addition to our original reference value (random (0,1) for the insertion and deletion evaluation criteria and include experiment results in Table 5,6 in Appendix L.
>
> We validate that for zero-baseline SHAP/IG, they perform relatively better when the reference value is smaller (in the range of 0 - 0.5), and perform significantly worse when the reference value is 0.75, 1, and random(0,1). On the other hand, our Greedy-AS’s performance is much stable across different reference values, which we conjecture that it is due to the fact that Greedy-AS does not assume any reference values. Our main evaluation is when the reference value is set to random (0,1), where zero-baseline SHAP/IG performs considerably worse than our method (which we conjecture is due to the fact that their method is tailored to the reference value zero). Another empirical evidence is that Expected Gradient (which is IG averaged over random reference points) performs much more stably across different reference values and also performs better when the reference value is set to random(0,1). We believe the added experiments show that our method is more stable across different reference values, and thus leads to better performance than IG. We include the intuition why our method performs decently well across different reference values in the next question response, and we included a simple theoretical analysis for explanations that are biased to the reference value in Appendix A.1.
>
> Q3: Greedy-AS does better on the benchmark that inspired it, but why does it do better on the removal-based methods? In some sense, this is exactly what methods like SHAP should be optimizing for, so the biggest drawback to this result (which is a good result for your method!) is that I don't have an intuition for why it should be the case. I'd also be interested to see a comparison with a method like Expected Gradients (Erion et al 2019, Sturmfels et al 2020), which averages IG-type explanations over multiple references by default.
>
> A3: Consider the Insertion score first, our Greedy-AS considers features that makes the prediction most robust to “adversarial perturbations’’ when anchored, and thus the prediction will still be robust to “any arbitrary perturbations (including different removal techniques)’’ when these features are anchored, since “adversarial perturbation’’ is the worst case of  “any arbitrary perturbations’’. In this viewpoint, our method can be seen as optimizing a lower bound of the Insertion score. Similarly, our method optimizes an upper bound of the Deletion score. The reason why SHAP and IG performs inferior in some cases is due to the mismatch of the reference value discussed above, and we included Expected Gradient in our revision.
>
> Q4: I'd also be interested in seeing the SHAP results on text. I think it may do a lot better in that case.
>
> A4: We provide an additional set of experiments on text classification with more comprehensive quantitative and qualitative results in revision Appendix M and N. In the experiments, we do observe SHAP yields strong performance on Insertion and Deletion criteria with zero baseline, while relatively suffering degraded performance under evaluations with random-baseline and our proposed criteria, which is also observed in the image examples in Appendix L. We believe such phenomenon could be attributed to the bias introduced by SHAP when operationalizing feature-removal with some selected reference value.

---

> ### Author Response · Authors · 2020-11-20
> **Response to Reviewer 2 (part 3)**
>
> Q5: IG is one of the best performers in the Table 1/2 benchmarks, but is not used in Figures 4 and 5.
>
> A5: We have included the results of IG on text classification in the revision Appendix M and N. Qualitatively on the text classification task, we observe that the top 1 to 2 keywords selected by IG matches human intuition well (the keywords appear to be important for predicting the ground truth class), while its top 3 to 5 keywords are more prone to be stop words. For example, in Figure 18 (b) where the ground truth class is “sports”, the top 5 keywords (from most relevant to less relevant) selected by IG are “Ronaldinho”, “why”, “did”, “they” and “them”; whereas our method selects “Ronaldinho”, “favorite”, “kaka”, “players”, and “switched”. We observe that the keywords selected by Greedy seem to have stronger relations to the ground truth class. We have expanded our original user study on text classification results (in Appendix O) to include more baselines to further justify the effectiveness of our method.
>
> In figure 5, as our original goal is to demonstrate how the proposed method could be adapted to provide targeted explanations, we did not include IG which is not designed to handle targeted explanations for comparison. We instead compare and discuss our method with several other existing explanations that have been discussed under similar context in Appendix K.
>
> Q6: I think there's not quite enough justification for the benchmark aside from "other methods are biased by the reference." However, there's not much empirical or theoretical basis for believing the use of a reference is intrinsically bad.
>
> A6: We have added a simple theoretical analysis of bias for reference value in Appendix A to justify the intuition of why adopting reference value might fail in different cases. Additional empirical results in Appendix L also demonstrate how the use of different reference values could strongly affect the evaluation results.
>
> In Appendix A.1, we have added theoretical analysis to explain why existing baselines are problematic in certain cases. We discussed one caveat for using reference value evaluations -- if the feature value equals the reference value, the feature will not be deemed as important (even if it is the key to prediction). In Appendix A.2, we added theoretical analysis that when a feature is equal to the replaced reference value, no matter how important it actually is, it will not contribute to the deletion score nor the insertion score. However, the lower the deletion score is better, and the higher the insertion score is better, and choosing an important feature that corresponds to the reference value will inevitably not improve the deletion score and insertion score. The reference values such as blurring out or adding noise may still make the original features unchanged (such as when the main part of the image is already blurred, blurring the image will not change the image, and thus the main part of the image is biased to blurred baseline).
>
> Q7: Overall, having more non-image experiments, and associated discussion, could be helpful.
>
> A7: In the revision, we have included a more comprehensive set of experiments on text classification with quantitative and qualitative results in Appendix M and N. We observe similar trends as in image experiments that our method has better performances on the proposed criteria and the removal-based criteria with random baseline. While SHAP achieves the strongest performance on the removal-based criteria with zero baseline, its performance decreases under criteria with mismatching baseline value. The results on text classification serve as additional empirical observations that reference value based explanation might fail when evaluated under criteria with mismatching baselines.

---

> ### Author Response · Authors · 2020-11-20
> **Response to Reviewer 2 (part 4)**
>
> Q8: Can you add some discussion of the runtime?
>
> A8: We provide in Appendix P an initial study on the runtime analysis for different methods. We show the average runtime (wall clock time) of different methods for computing explanation for a single MNIST image. For Greedy and Greedy-AS, we show the time needed to compute the top-20% relevant features.
>
> We see that LOO has the least running time since its computation can be done with a single batch forward pass to the model. Gradient-based methods are also fast since they only need constant times of forwarding and backward passes of the model. Optimization-based methods like BBMP and CFX relatively take a longer time as they require multiple steps of gradient descent. SHAP as well requires longer runtime as it depends on sampling and solving a kernel regression. Comparing to existing explanations, our method indeed is not as fast since we require running PGD for multiple candidate sets to evaluate their robustness in each greedy step. We could however speed up the process by leveraging GPU to parallelly perform PGD on multiple candidates simultaneously and further reduce the computational time by relaxing PGD constraints such as performing fewer descent steps (with the trade-off of less precise robustness value) and selecting multiple features at each greedy step. We note that the time reported is measured with a single GPU, and further speed-up is available by parallelizing the computation over multiple GPUs. We will include a detailed runtime comparison over different methods in the final revision.

---

> > ### Comment · AnonReviewer2 · 2020-11-24
> > **Thanks for a comprehensive response!**
> >
> > Thanks to the authors for a very comprehensive response and detailed analysis on all of my points. My major concerns have been addressed well. I think there's still a lot of interesting discussion to be had around the use of references as well as the type and size of perturbations. However, it's clear to me that this paper, especially with the newly added material, is a thoughtful exploration of such issues, and I believe it is well above the acceptance threshold.

---

### Author Response · Authors · 2020-11-20
**Initial General Response**

We thank all reviewers for their helpful and constructive comments. We posted our initial response a bit late since there were 4 reviewers with a lot of great questions, and we have worked on incorporating the suggestions into our initial revision and will continue to address all the remaining concerns by the end of the rebuttal period. Please let us know if there are any more follow-up questions, and we will try our best to respond. In the initial revision, we have (1) expanded the comparison and discussion on text dataset with both quantitative and qualitative results, (2) conducted a careful analysis to discuss the potential caveats (bias of reference value issue) of evaluations and explanation methods that rely on adopting reference value (by theoretic formulation in appendix A and by empirical studies of different reference value for the insertion score and deletion score in appendix L and M), (3) added Expected Gradient and random explanation as additional baselines, (4) conducted an initial runtime analysis on different explanation methods, (5) added clarification to the user study and compared more baseline methods, (6) conducted an overall polishing of the paper.

By the end of the rebuttal period, we aim to further incorporate (1) an expanded set of user studies involving more participants as suggested by Reviewer 4, and (2) more evaluation measures as suggested by Reviewer 3. We thank the reviewers for their patience while we gather additional experimental results.

---

> ### Author Response · Authors · 2020-11-24
> **General Response Update**
>
> We thank all reviewers for carefully reviewing our paper and providing us many constructive feedbacks. We have further incorporated several updates in our revision and have hopefully addressed all the concerns raised. In brief, we have (1) included an expanded set of user study involving a total of 30 users to validate the results on text classification, (2) conducted a sensitivity analysis as additional measurement for evaluating explanations, and (3) included more discussion to related literature in the paper.
>
> In the meantime, we welcome more follow-up questions and look forward to more discussions.

---

### Decision · Program_Chairs · 2021-01-07
**Final Decision**

**Decision:**

Accept (Poster)

**Comment:**

The authors introduce new evaluation criteria and methods for identifying salient features: rather than earlier approaches which attempt to 'remove' or marginalize out features in various ways, here they consider robustness analysis with small adversarial perturbations in an Lp ball.  For text classification, a user study is included which is appreciated.

In discussion, the authors addressed many points and all reviewers converged to recommend acceptance.

A couple of points could be discussed further if space permits:
the impact of type of perturbation employed; and
the connection between optimizing for adversarial robustness and optimizing for insertion/deletion criteria.